# Review article: Physical Vulnerability Database for Critical Infrastructure Hazard Risk Assessments – A systematic review and data collection

Sadhana Nirandjan[1], Elco E. Koks[1,2], Mengqi Ye[1], Raghav Pant[2], Kees C.H. Van Ginkel[1,3], Jeroen C. J. H. Aerts[1], and Philip J. Ward[1]

[1]Institute for Environmental Studies (IVM), Vrije Universiteit Amsterdam, 1081HV Amsterdam, The Netherlands
[2]Environmental Change Institute, University of Oxford, Oxford OX1 3QY, United Kingdom
[3]Inland Water Systems, Deltares, 2629HV Delft, the Netherlands

*Correspondence to*: Sadhana Nirandjan (sadhana.nirandjan@vu.nl)

**Abstract.** Critical infrastructure (CI) is exposed to natural hazards that may lead to the devastation of these infrastructures and burden society with the indirect consequences that stem from this. Fragility and vulnerability curves, which quantify the likelihood of a certain damage state and the level of damage of an element under varying hazard intensities, play a crucial role in comprehending, evaluating and mitigating the damages posed by natural hazards to these infrastructures. To date, however, these curves for CI are distributed across literature instead of being accessible through a centralized dataset. This study, through a systematic literature review, synthesises the state-of-the-art of fragility and vulnerability curves for CI assets of energy, transport, water, waste, telecoms, health and education in context of natural hazards and offers a unique physical vulnerability database. The publicly available centralized database that contains over 1,510 curves can directly be used as input for risk assessment studies that evaluate the potential physical damages to assets due to flooding, earthquakes, windstorms and landslides. The literature review highlights that vulnerability development has mainly focused on earthquake curves for a wide range of infrastructure types. Windstorms have the second largest share in the database, but are especially limited to energy curves. While all CI systems require more vulnerability research, additional efforts are needed for telecommunication which is largely underrepresented in our database.

## 1 Introduction

Globally, critical infrastructure (CI), constituting of energy, transport, water, waste, telecoms, health and education systems, are increasingly at risk to natural hazards (Izaguirre et al., 2021; Stewart and Rosowsky, 2022; Verschuur et al., 2023). This is driven by both a growing demand for infrastructure associated with socio-economic development, and an observed and projected increase in the frequency and intensity of climate extremes (IPCC, 2022). The level of vulnerability of CI to natural hazards is a key determinant for understanding, assessing and reducing natural hazard-induced risks to these infrastructures (Schneiderbauer et al., 2017). Indeed, the United Nation's Sendai Framework for Disaster Risk Reduction underscores that

enhanced work is needed to reduce vulnerabilities, and that freely available and accessible vulnerability information should be promoted for effective risk management (UNDRR, 2015).

Vulnerability is generally defined as 'the conditions determined by physical, social, economic and environmental factors or processes which increase the susceptibility of an individual, a community, assets or systems to the impacts of hazards' (UNDRR, 2022). When assessing the physical damage to a structural element due to direct contact with a hazard, a common approach to account for vulnerability of assets is through the use of "vulnerability curves" (Meyer et al., 2013). These curves relate given levels of a hazard intensity measure (e.g., flood inundation depth, wind speed) to the potential physical damage of an asset. The potential damage can either be expressed in absolute monetary terms, or in relative numbers that are often referred to as the mean damage ratio (MDR), which is commonly expressed as the ratio of the expected repair cost to the replacement costs of a structure (WBG, 2019). In the latter case, the MDR is then multiplied by a cost feature to obtain the potential damage for a given hazard intensity level. In an alternative approach, "fragility curves" describe the probability of reaching or exceeding a (number of) damage state(s) for a given hazard intensity measure (Douglas, 2007). A damage state describes the level of damage (e.g., 'Extensive') and is usually explained in a qualitative and descriptive way (e.g., major cracks in walls). The development of fragility curves is particularly emphasized within the earthquake community (Douglas, 2007), whereas the flood community tends to focus more on the development of vulnerability curves (Meyer et al., 2013).

While researchers have made significant progress in the development of fragility and vulnerability curves focusing on physical damages of different CI assets due to various natural hazards, no study has yet combined these curves into one extensive (multiple hazard) database for CI. Existing fragility and vulnerability curves are mostly distributed across peer-reviewed articles, (technical) reports, manuals and other literature rather than being centralized in one dataset. The limited number of existing open-access databases predominantly focus on structural damages to types of (residential) buildings. For example, the earthquake risk assessment initiatives Global Earthquake Model (GEM) and the Comprehensive Approach to Probabilistic Risk Assessment (CAPRA) platform support an extensive database containing functions for a range of building types (Cardona et al., 2012; Yepes-Estrada et al., 2016). Other, non-public databases are, for example, the Multi-Coloured Handbook (Penning-Rowsell et al., 2013) and fragility and vulnerability curves developed by the insurance industry. However, even within these non-public databases, CI is often inadequately represented. Furthermore, curves are often presented in a format that restrict researchers from directly using them. For example, Habermann and Hedel (2018) review a range of vulnerability functions for transport infrastructure exposed to fluvial floods and wildfires, but only present visualisations rather than underlying curve equations or values. A consistent overview of existing curves and an associated centralized, freely accessible database are lacking, despite the benefits they would provide to the disaster risk community. These resources would enable them to perform risk assessments supported by well-informed decisions based on the current state of the fragility and vulnerability literature.

As such, this study aims to develop an open-access CI vulnerability database for a selection of hazards (i.e., flooding, earthquakes, windstorms and landslides) by reviewing and extracting data from 95 studies across peer-reviewed and grey literature. The database comprises fragility and vulnerability curves, which have been normalised and standardised to be useful in a comparable way. The results of this study can be used as input for risk assessment studies that identify the natural hazard risk in terms of physical damages for a range of CI types. Moreover, we aim to identify gaps in the current state of literature in order to understand the aspects of vulnerability on which future research should focus.

## 2 Data and approach

The CI vulnerability database developed in this research builds on the CI categorization presented by Nirandjan et al. (2022), where we use seven overarching CI systems, namely: energy, transportation, telecommunication, water, waste, health, and education. Within each CI system, an extensive set of infrastructure asset types is included. The remainder of this section explains the search and screening procedure of the literature, and the setup of the database.

### 2.1 Literature review

The schematic workflow for the literature search, screening and final selection of articles for the systematic literature review on the CI vulnerability to flooding, earthquakes, windstorms and landslides is summarized in Fig. 1. The hazards were chosen based on their widespread occurrence, significant potential for damage to CI and historical evidence of their impact on communities. Our review is not restricted to peer-reviewed academic articles as curves are also published in 'grey literature', such as research reports released by governments or engineering firms. We therefore use Google Scholar as search engine that is not limited to academic literature in order to minimize the possibility of excluding relevant information within our research scope. We conducted a literature search and screening over the period January 2022 to March 2023 by systematically using combinations of keywords on the general concept of hazards, critical infrastructure and vulnerability (Table 1). The literature search yielded 2,590,003 initial records, gathered from 125 search term syntaxes listed in Appendix A. It became apparent that a substantial number of papers did not address CI vulnerability in context of natural hazards. As a result, we decided to select the first 250 records for each search term syntax, totalling 31,250 records for the screening procedure.

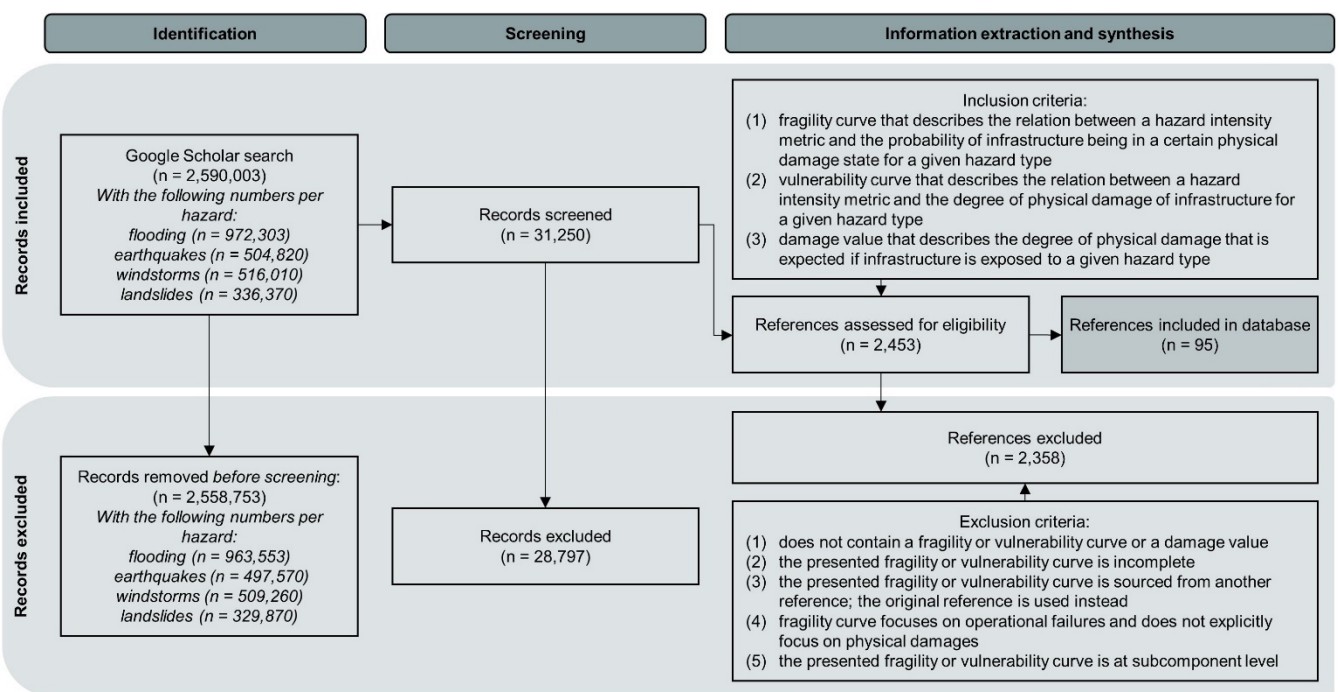

**Figure 1: Schematic display of the workflow, including the literature search, screening and final selection of articles for review, adapted from Moher et al. (2009).**

The records were screened for eligibility using three inclusion criteria which assess whether a record provides quantitative information about the vulnerability of a CI to potential damage from flooding, earthquakes, windstorms or landslides. To be included within the database, the literature must contain at least one of the following: (1) fragility curve that describes the relation between a hazard intensity measure and the damage probability, i.e., probability of infrastructure being in a certain physical damage state for a given hazard type, (2) vulnerability curve that describes the relation between a hazard intensity measure and the degree of physical damage of infrastructure for a given hazard type, and/or a (3) damage value that describes the degree of physical damage that is expected if infrastructure is exposed to a given hazard type. This is challenging since many papers broadly discuss vulnerability aspects of CI, but often do not present specific curves or provide them only in an incomplete way (e.g., figure given but axis missing). If multiple records present the same curves, we only include the original source reference. We also excluded records that describe the probability of an asset failing to operate rather than the damage probability of being in a certain physical damage state, as we confine the scope of this research to fragility curves that specifically involve the physical damage (see inclusion criteria 1). Note that we exclude curves at subcomponent level (e.g., circuit switcher), but do include them if they are at asset- or system level. Furthermore, we limited our literature review to research written in English or Dutch. However, we did not limit the search window and the geographical scope of the study and are thus still able to provide insight into curves in various contexts.

**Table 1: Keywords for the three general concepts (i.e., hazards, critical infrastructure, vulnerability) of the literature search. The keywords in italics are the infrastructure asset types included in this study.**

| General concept | Keywords |
|---|---|
| Hazards | natural disaster, natural hazard, flooding, flood, earthquake, tropical cyclone, cyclone, hurricane, windstorm, storm, wind, landslide |
| Critical infrastructure | critical infrastructure, lifeline, energy, power, transportation, telecommunication, water, waste, health, education, *power plant, substation, power tower, power pole, cable, power line, railway, roads, airports, runway, communication tower, mast, water tower, water well, water works, waste transfer station, wastewater treatment plant, health facility, hospital, school* |
| Vulnerability | vulnerability curve, fragility curve, depth-damage function, depth-damage curve |


The procedure resulted in 95 references with useful information for the database. Specifically for flood, hurricane and earthquake risk in the United States (US) the Federal Emergency Management Agency (FEMA, 2013, 2021a, 2020) developed technical manuals that contain curves for infrastructure. The large contribution of FEMA to our database is apparent: the US has the highest number of curves, with 195 (24.3%) sets of fragility and vulnerability curves stemming from this source.

Another source for cross-hazard and cross-infrastructure curves is Miyamoto International (MI, 2019) that present curves for application at the global scale. We would like to stress that our database does not encompass all types of infrastructure. There is already vast literature available for limited infrastructure types. For bridges, for example, 224 bridge damage curves for 28 primary bridge types are offered by FEMA (2020) and a dedicated review is provided by Muntasir Billah and Shahria Alam (2015). Moreover, retrieving curves is labour-intensive. Instead, our focus was on delivering as comprehensive a review as

possible for the infrastructure types as presented in table 1.

**2.2 Setup of the CI vulnerability database**

The database is available through Zenodo (see *Data availability*) and consists of three spreadsheets: Table D1, D2 and D3. For setting up the database, we systematically assess the literature on hazard, exposure and vulnerability characteristics that are listed below. In addition, for each curve we indicate a number of details regarding reliability and reference purposes. Table D1

summarizes these aspects of the curves.

Hazard
- *Hazard type.* We indicate the hazard type the curve represents, including: flooding (coastal, river, and surface), earthquakes, windstorms (tropical and extra-topical) and landslides (rainfall- and earthquake triggered).
- *Intensity measure.* We specify which hazard intensity measure is used.

Exposed element
- *Infrastructure description.* We specify the infrastructure asset type to which the curve is applicable.
- *Additional characteristics.* We elaborate on any characteristics of the infrastructure asset type that should specifically

be mentioned if these characteristics are fundamental for the vulnerability of a given infrastructure asset to a specific hazard type (e.g. type of construction material, installation height of essential equipment, and inventory). We also provide environmental characteristics that influence the vulnerability of an exposed element (e.g., corrosive soil conditions) if specified in the source. Please note that we also provide details if a curve incorporated conditions that are sustained from a previous hazard.


Vulnerability details

- *Fragility and/or vulnerability.* We indicate whether a fragility curve or a vulnerability curve is provided (or both).
- *Characteristics of the curve.* We indicate whether a given curve is continuous (i.e., joined discrete points) or discontinuous, and whether the damage measure is expressed in absolute or relative terms. Note that fragility curves
are always expressed in relative terms (i.e., relative probability).
- *Damage states.* In case of a fragility curve, we indicate the number of damage states considered and, if provided in the source, the associated level of structural damage.
- *Cost feature.* We indicate whether a cost feature is provided that can be used in combination with the curve. This cost feature is commonly based on either replacement costs (i.e., the amount, which is based on market values, needed to
replace an object with a comparable object) or reconstruction costs (i.e., the amount needed to rebuild an object to its original state at the same location).
- *Uncertainty range.* We indicate whether an uncertainty range is provided that can be used to quantitatively estimate the bandwidth of modelled damages. The uncertainty range can either be in the form of an upper and lower boundary for a curve or a range in cost features.
- *Derivation method.* We specify the method that is applied to derive the curve, differentiating between the following methods: analytical, empirical, expert opinion and hybrid. The analytical approach relies on numerical models or analytical formulations, the empirical approach on post-hazard damage data, the expert-opinion approach on the subjective opinion of a group of experts, and the hybrid approach is based on a combination of two or more of the aforementioned approaches (D'Ayala et al., 2015b).
- *Geographical application*. We indicate the region for which the curve is developed.

Source details

- *Source type.* We indicate the source type from which the curve is retrieved. This may be peer-reviewed or grey literature. If the latter is the case, we specify whether the source type is a technical manual, report, conference
proceeding, or another type of source.
- *Readily available.* We specify whether the curve was readily available, meaning that the original source provided datapoints, parameters or a formula to reconstruct the curve. If these were not provided, we made a best estimate

based on the figure to replicate the curve.

- *ID number.* Each vulnerability curve and set of fragility curves is provided with a unique identifier.
- *Original ID number.* If the original source labelled the curve (e.g., curve number 1), we provide this label in our overview to aid reference purposes.

The final collection of fragility and vulnerability curves for flooding, earthquakes, windstorms and landslides is provided in Table D2. To consistently report the curves in the database, we pursued the following:

- Curves were not always provided in the same units (e.g., for inundation depth, curves were available in meters and in feet). We therefore converted the fragility and vulnerability curves to one unique unit for each intensity measure (e.g., meters for inundation depth).
- All curves are presented as relative functions in the database. If vulnerability curves were originally provided as absolute functions, we converted it into functions in a relative way that ranges between 0 (no damage) to 1 (maximum damage).
- If the original source only provided a figure instead of actual numerical values, parameters or an equation for the reproduction of the curve, we estimated the numerical values of the curve. If the original source only provides numerical values for certain intensity levels, we interpolated linearly between known values. In Table S2, the estimated values are highlighted in yellow and the interpolated values in green.

Furthermore, complementary to the curve database, Table D3 contains cost numbers that can be used in combination with the curves for the estimation of potential damages, if provided in the original source from which the curve is retrieved. We indicate the infrastructure asset type, the amount and potential bandwidths, the geographical application, and on what information it is based (e.g., replacement, construction or repair costs). We converted the cost values to 2020 as reference year using the Consumer Price Index provided by the World Bank Group (WBG, 2023). For consistency purposes, the ID numbers given throughout this paper match with the summary table (Table D1), the curves in Table D2 and cost numbers in Table D3.

### 2.3 Standardisation of the fragility and vulnerability curves

While vulnerability curves directly allow for estimating damage on the basis of a hazard intensity measure (e.g., flood depth), fragility curves require a procedure that entails the transformation of these curves to vulnerability curves so that a relative cost is given for each hazard intensity measure level. Damage-to-loss models, also known as damage-to-impact or consequence models, act as a crucial link between fragility and vulnerability curves by relating physical damage with a damage or loss metric (Martins et al., 2016; Yepes-Estrada et al., 2016; Gentile et al., 2022). A review of these damage-to-loss models, however, is outside the scope of this study. The transformation can be achieved based on the following: (1) the complementary cumulative cost distribution for a given damage state $E(C|ds_i)$, and (2) the probability of being in a certain damage state for

a given intensity level $P(ds_i|im)$. The cumulative distribution of cost given an intensity level $E(C|im)$, also referred to as the MDR (WBG, 2019) or the compounded damage ratio (FEMA, 2020), is computed as follows (D'Ayala et al., 2015a; WBG, 2019):

$$E(C|im) = \sum_{i=1}^{n} E(C|ds_i) * P(ds_i|im) \qquad (1)$$

Here, the damage state 'None' that expresses no damage to an element is not included in the number of damage states $n$ that are considered in the summation. If a range of $E(C|ds_i)$ is given, we use this this range to derive upper and lower bounds of the vulnerability curve. If not, we calculate the variance for each intensity level, which is derived as follows (D'Ayala et al., 2015a; WBG, 2019):

$$var(C|im) = \sum_{i=1}^{n} (E(C|ds_i) - E(c|im))^2 * P(ds_i|im) \qquad (2)$$

Unfortunately, a complementary cost distribution for a damage state is not always provided in the original source. In this review, we do not fill the gaps based on assumptions, but provide the vulnerability information as it is. We therefore only apply the transformation procedure if the cost distribution element is actually given. If not, we only include the curve in our database as a fragility curve. Figure 2 illustrates generic examples of fragility curves for various damage states and a vulnerability curve.

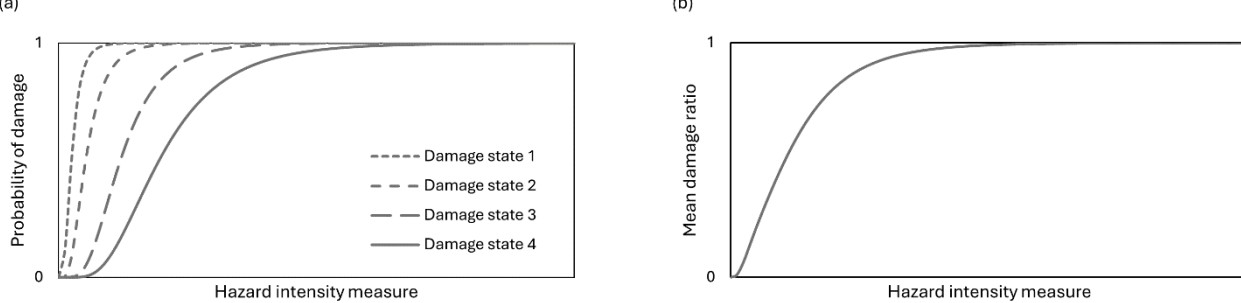

**Figure 2: Representations for fragility curves in panel A and a vulnerability curve in panel B. The fragility curves for four damage states are illustrated, which could be categorized as follows: $ds_1$ = Slight Damage, $ds_2$ = Moderate Damage, $ds_3$ = Extensive Damage, $ds_4$ = Complete Damage.**

**3 Review of CI vulnerability literature per hazard type**

This section summarizes the fragility and vulnerability curves per overarching CI system, grouped in four hazard subsections (3.1 to 3.4). Figure 3 indicates the number of unique curves found in existing literature as well as the number of countries that are represented by these curves for the reviewed infrastructure-hazard combinations. Moreover, we indicate the available curve types for each infrastructure-hazard combination. The findings are provided for curves that represent infrastructure at system-

level (i.e., the overarching CI systems) and asset-level (i.e., the assets that are part of the CI system). In consideration of the review's length, we have chosen to not to delve into detailed discussions of all hazard, exposure and vulnerability characteristics for each curve. Instead, we focused on offering a concise overview of the current vulnerability literature in this section, whilst a complete overview of the hazard, exposure and vulnerability characteristics as discussed in Section 2.2 can be found in Table D1 of our dataset.

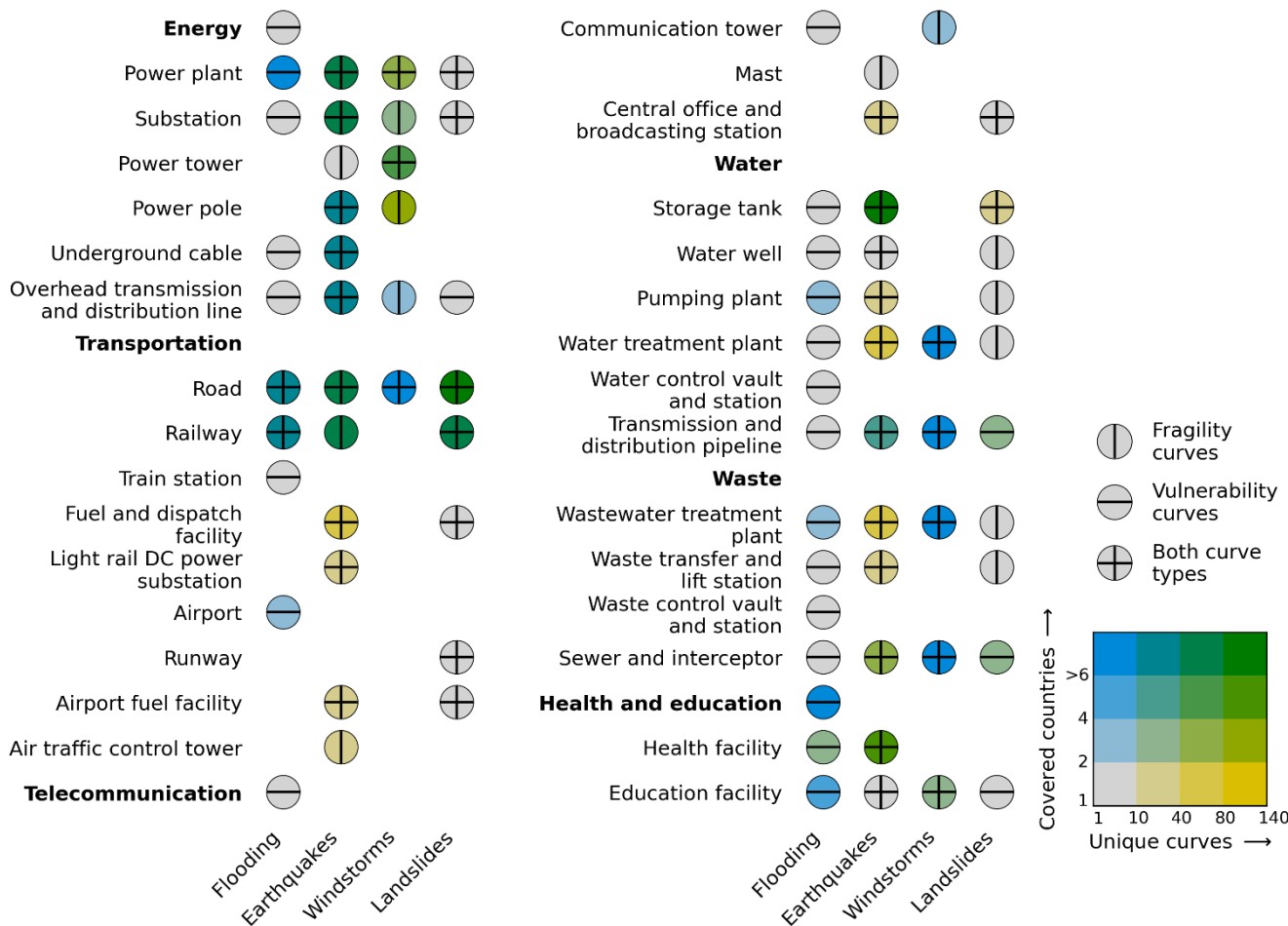

**Figure 3: Summary of findings across the reviewed infrastructure and hazard types. The abundancy of curves (i.e., the number of unique curves) and the geographical coverage (i.e., the number of countries that are covered by these curves) are highlighted by colour for the infrastructure-hazard combinations. Additionally, the curve type (i.e., fragility, vulnerability or both) is also indicated for the infrastructure-hazard combinations. Furthermore, the infrastructure highlighted in bold represent the overarching CI systems for which generalized curves are available for flooding in particular.**

### 3.1 Flooding

#### 3.1.1 Energy

FEMA (2013) developed depth-damage curves (i.e., vulnerability curves that relate the flood inundation depth to the potential physical damage), for power plants with varying capacities for the US, which are assumed to be identical in shape [F1.1-3] with replacement costs varying depending on the capacity of the power plant. MI (2019) assume that the vulnerability of coal, gas or oil-based thermal plants [F1.4] is similar to the vulnerability curves for power plants of FEMA (2013). Wind farms are not vulnerable to flooding according to MI (2019) [F1.5], whereas Vanneuville et al. (2006) assumes that flood damages to wind turbines can reach up to 712,000 EUR/unit [F1.6]. For risk assessments in Schleswig-Holstein, Germany, a relatively low vulnerability is assumed for wind turbines [F1.7]; up to 3.5% of the value of a unit (Meyer and Messner, 2005). This is based on a national flood damage database (HOWAS) and input of experts to develop curves that represent the regional conditions.

FEMA (2013) considers transmission (138-765 kV) and sub-transmission (34.5-161 kV) substations, categorized as small (low voltage: 34.5-150 kV), medium (medium voltage: 150-350 kV), and large (high voltage: >350 kV). The shapes of the vulnerability curves for the three categories of substations are identical [F2.1-3]. The general assumptions on which the curves are developed are that: electrical switch gear is located at a height of 0.91 m above ground level; damage to the control room starts at the onset of the flood and is maximized when reaching a water level of 2.13 m; and electrical components (e.g., cabling, transformers, and switchgear) are also damaged.

FEMA (2013) developed three vulnerability curves for the distribution circuit, which we differentiated into curves for the underground transmission and distribution (T&D) system (i.e., cables) [F5.1] and the overhead T&D system (i.e., power (minor) lines) [F6.1-2]. Underground and overhead infrastructure are assumed to stay unharmed due to inundation, while there is a low vulnerability expected at the end of buried cables. Furthermore, Kok et al. (2005) provide a generalized depth-damage curve for the estimation of flood risk to energy systems within the Netherlands [F6.3]. We did not find curves for power towers and poles.

#### 3.1.2 Transportation

Huizinga (2007) developed a set of depth-damage curves for diverse land use classes including transport infrastructure, initially for the European Union (EU) and later generalized worldwide (Huizinga et al., 2017). These curves differentiate between the land use classes 'transport' and 'infrastructure'. 'Transport' is defined as 'transport facilities', which seems to refer to transport terminals such as railway stations, ports, and airports. 'Infrastructure' is defined as physical damage to "roads and railways as a result of contact with (fast flowing) water" (Huizinga et al., 2017). Although the latter curve is widely applied to diverse infrastructure classes (e.g., Albano et al., 2017; Dottori et al., 2023), the background document (Huizinga, 2007) shows that it

is explicitly derived for road infrastructure [F7.1-3]. For use in asset-based models that require highly spatially detailed infrastructure data rather than generalized land use classes, Van Ginkel et al. (2021) developed a new set of depth-damage curves specifically for roads in the EU and tailored them to six different road types in OpenStreetMap (OSM), correcting for the number of lanes [F7.4-9].

McKenna et al. (2021) provides analytically derived fragility and vulnerability curves [F7.14-15] for granular highway embankments. They use the Water Intensity Measure (WIM) as intensity measure, which describes the proportion of the embankment height that would be considered saturated if exposed to moisture ingress due to flooding. Additionally, they also assess the impact of scouring using a scouring depth of 0.5 and 3 m as lower and upper boundary, respectively, whilst the raised groundwater level was maintained. Their study shows that higher damages are expected with increasing moisture ingress and scour depths.

Kok et al. (2005) developed a road depth-damage curve [F7.10] based on a limited amount of damage data and expert judgement, which is one of the curves originally used for a standard method for flood damage evaluation in the Netherlands and has been adopted for risk assessments in Belgium [F7.11] (Vanneuville et al., 2006). An updated version of the curve assumes a lower vulnerability for water depths under 25 cm, an increasing vulnerability thereafter due to electric accessories being damaged, followed by a less steep slope as additional water is not expected to result in significant additional damage [F7.12] (de Bruijn et al., 2015). The Rhine Atlas damage model (RAM; ICPR, 2001) involves five depth-damage curves using damage records from Germany and expert judgement (Bubeck and de Moel, 2010; Bubeck et al., 2011), of which a generic 'traffic' curve is developed for applications to the infrastructure sector [F7.13] (Kellermann et al., 2015).

A depth-damage curve for Austrian railways is presented by Kellermann et al. (2015) [F8.1], with applications of the RAilway Infrastructure Loss (RAIL) model at the local scale (Kellermann et al., 2015), regional scale (Kellermann et al., 2016), and European scale (Bubeck et al., 2019). The road curves of Kok et al. (2005), Vanneuville et al. (2006), de Bruijn et al. (2015) and Huizinga et al. (2017) are also applied to railways [F8.2-7]. These curves, however, are a generalized representation for linear infrastructure, whereas the Kellermann et al. (2015) curve is explicitly developed for the Austrian Northern Railway line and other railways with the same structural characteristics. Furthermore, Vanneuville et al. (2006) assume that the generic curve for the industry sector can be applied to train stations in Belgium [F8.8].

Tsubaki et al. (2016) explains that railway damage commonly occurs due to floodwater overtopping leading to scouring of the ballast and embankment upon which the trail tracks are built. Railway overtopping damage begins with ballast scour and progresses to embankment scour. They therefore developed fragility curves for ballast scour damage, embankment fill scour damage and a combination of both damage conditions [F8.9-11] using damage records of flood events for single-track railways in Japan.

A depth-damage curve for airports and the associated costs is presented by Kok et al. (2005) [F9.1]. The curve, however, is a generalized curve that is also used for land uses with the occupation agriculture and recreation. De Bruijn et al. (2015) propose a depth-damage curve that assumes an overall lower vulnerability of airports [F9.2] instead of depth-damage curve [F9.1]. Vanneuville et al. (2006) present a generalized depth-damage curve for industry that can be applied to airports [F9.3], which assumes (1) a lower overall vulnerability compared to depth-damage curve [F9.1]; (2) a slightly lower vulnerability between a water depth of 0.4-2 m compared to depth-damage curve [F9.2]; and (3) the maximum damage being reached with a water depth of 4 m.

### 3.1.3 Telecommunication

Kok et al. (2005) provide a depth-damage curve for the estimation of flood risk to communication systems within the Netherlands [F12.1]. The curve is generalized and not developed for specific structures within the communication system. For flood risk assessments in Belgium, Vanneuville et al. (2006) propose a depth-damage curve for communication towers [F10.1].

### 3.1.4 Water

FEMA (2013) provides vulnerability curves and reconstruction costs for (potable) water system facilities, including water treatment plants, pumping stations, storage tanks and wells.

Five depth-damage curves were developed by FEMA (2013) for storage tanks [F13.1-5] that typically have a capacity of 1.9-7.6 million L/day (FEMA, 2021b), with varying elevation levels (i.e., at ground level, elevated, or below ground level), and construction materials (i.e., wood, steel, or concrete). Storage tanks at ground level and elevated are assumed not to be vulnerable to flooding. For storage tanks at ground level it is assumed that the water level in the tank exceeds the flood depth, thus preventing the storage tank from floating. For elevated storage tanks it is assumed that the tank foundations are not damaged. Storage tanks that are situated below ground level are assumed to be vulnerable to flooding, with the underlying assumption that the tank vent is 0.91 m above ground level, and that clean-up will be required after flooding.

A number of depth-damage curves are developed by FEMA (2013) for water treatment plants (WTP) [F14.1-10], which are generally composed of a number of interconnected pipes, basins and channels required for physical and chemical processes to improve water quality. In general, the curves for open WTPs follow the same shape regardless of the capacity, as do the ones developed for closed and pressurized WTPs. Here, the depth-damage curve developed for open WTPs assume a higher vulnerability compared to the closed and pressurized WTPs. Also, they developed a depth-damage curve for water wells that typically have a capacity between 3.8 and 18.9 million L/day [F15.1], under the assumption that electrical equipment and well openings are 0.91 m above ground level and that a well is not permanently contaminated after flooding. According to FEMA (2013), transmission pipelines for potable water are not expected to be harmed due to flooding [F16.1-3].

Pumping plants are typically composed of a building, one or more pumps, electrical equipment and occasionally with backup power systems. FEMA (2013) developed depth-damage curves for pumping plants on the basis of elevation level and capacity [F17.1-4], with the first being the determinant of the vulnerability level. For pumping plants below ground level, it is assumed that the entrance is 0.91 m above ground level. Flood water starts entering the pumping plant once this critical height is exceeded, hereby damaging electrical equipment that is assumed to be below ground level. In contrast, the depth-damage curve for pumping stations above ground level propagates gradually. Kok et al. (2005) present a depth-damage curve for pumping stations that can be used in combination with a cost value to estimate the direct physical damage to pumping stations in the Netherlands [F17.5]. The depth-damage curve is developed for pumping stations with a capacity of 518 L/d that are located in areas with a return period lower than 25 years. Furthermore, FEMA (2013) provides a depth-damage curve and associated reconstruction costs for control vaults and stations [F17.6]. They assume that the entrance is at ground level and that water can enter control vaults and stations, resulting in a damage of 40% of the reconstruction costs.

### 3.1.5 Waste

FEMA (2013) provides vulnerability curves for various waste assets. Infrastructure components of wastewater treatment plants (WWTPs) are similar to those described for WTPs, but with the addition of secondary treatment subcomponents. Depth-damage curves [F18.1-5] were developed by FEMA (2013) for three categories of WWTPs (small, medium and large, depending on capacity). The shape of the curve is similar for the three categories, whereby it is assumed that clean up, repair of small motors, buried conduits and transformers is required from the onset of the flooding. Clean up and major repair of electrical equipment is required when the flood inundation level exceeds 0.91 m. Kok et al. (2005) present a depth-damage curve for WWTPs and a cost value to estimate the direct physical damage to WWTPs in the Netherlands [F18.6].

FEMA (2013) defines three categories for the waste transmission system [F19.1-3], assuming that no to little damage is expected from submergence. Four depth-damage curves were developed by FEMA (2013) for lift stations, which are facilities to pressurize the waste system aiming to raise sewage over topographical rises [F20.2-5]. If such a lift station is disrupted, untreated sewage may spill out near the lift station, or flows back into a collection sewer system (FEMA, 2021b). Lift stations are classified based on capacity as either small (<38 million L/d), medium (38-189 million L/d), or large (>189 L/d), and whether the lift station is flood proof. The non-flood proof lift stations are assumed to be damaged up to 40% of the reconstruction costs by flood water, while flood-proof lift stations may experience a damage only up to 10%. FEMA (2013) provides a depth-damage curve and associated reconstruction costs for control vaults and stations [F20.1]. They assume that the entrance is at ground level and that water can enter control vaults and stations, resulting in a damage of 40% of the reconstruction costs.

### 3.1.6 Health & Education

Huizinga et al. (2017) developed depth-damage curves for the category 'commercial buildings', which also includes schools and hospitals. These curves are generated for Europe, North America, Central- and South America, Asia, Oceania, and at the global scale based on flood damage data and country-specific information [F21.1-6]. Kok et al. (2005) present a general depth-damage curve that is applied for companies and governmental buildings, including education institutions (e.g., universities) and social services (e.g., hospitals), in low-frequency flooded areas [F21.7]. De Bruijn et al. (2015) propose to refine the generalized depth-damage curve [F21.7] into three categories and present a specific curve for 'offices' [F21.8] that encompasses educational and health facilities as well. Vanneuville et al. (2006) presents a generalized curve for buildings that also includes school buildings [F21.10], and depth-damage curve [F9.3] for airports is applied to hospitals [21.9]. Compared to the curve for schools, hospitals are assumed to have a higher vulnerability.

Djordjević (2014) relates the flood depth to the absolute damage per m$^2$ for schools in the city of Taipei, Taiwan, by using the available literature in combination with field surveys and expert judgement [F21.11]. The same methodology is applied to develop depth-damage curves representing school and health facilities in the municipality of Châtelaillon-Plage, located at the Atlantic coast of France (Batica et al., 2018). Health facilities [F21.12] are assumed to be more vulnerable to floods compared to education facilities [F21.13], with the maximum damage being reached at a water depth of 3 m. FEMA (2013) provides curves for essential facilities, which includes both health and education facilities. More specifically, curves are available for hospitals (with varying capacities), medical clinics (e.g., clinics, labs, and blood banks), schools (i.e., primary/secondary schools), and colleges/universities (i.e., Community and State colleges, State and Private Universities). These curves are only accessible via their HAZUS software, and are therefore not included in our database.

### 3.2 Earthquakes

#### 3.2.1 Energy

For power plants, FEMA (2020) developed fragility curves using the level of ground motion expressed in Peak Ground Acceleration (PGA) as hazard intensity measure. Fragility curves are developed probabilistically using Boolean expressions that describe the relationship of subcomponents, resulting in sets of fragility curves [E1.1-4] for plants with varying capacity and structural design (i.e., unanchored and anchored components). For thermal plants, MI (2019) adjusted the FEMA (2020) fragility curves based on expert-opinion to represent the global higher vulnerability for unanchored and the lower vulnerability for anchored thermal plants [E1.5-6]. Hydropower plants are vulnerable to earthquakes and potential failure mechanisms include sliding or overturning of the dam, and structural failure of components (e.g. bottom outlets, gates, and spillways). MI (2019) assume that failure due to sliding results in complete destruction, and adapted the base-sliding curves for concrete gravity dams by Ghanaat et al. (2012) for the representation of hydropower plants at the global scale [E1.7-8]. Gautam and Rupakhety (2021) developed a set of fragility curves for hydropower systems [E1.15-16] in Nepal based on empirical evidence

from the 2015 Gorkha earthquake ($M_w$ 7.8). The fragility curves consider 'minor', 'moderate, and 'major' damage states, and are provided for two intensity measures: PGA, and Peak Ground Velocity (PGV). For solar farms, MI (2019) assumes that they contain light structures of which the members and connections may be vulnerable to earthquakes; therefore, FEMA (2020) curves for steel light-frame buildings could be used [E1.9-10].

Myers et al. (2012) apply an analytical approach to develop fragility curves for two wind turbines that are 80 m tall, but with different capacities, support tower geometry and steel grade [E1.11-12]. The fragility curves represent the probability of 'severe' damage, meaning turbines that are locally buckled and collapsed. Wind turbines have no redundancy in their structural design; if one section is sufficiently damaged, the entire structure may collapse (Myers et al., 2012). According to Nuta et al. (2011), a wind turbine can be considered as a complete loss after the first buckle is created. Martín del Campo et al. (2021) developed fragility curves for wind turbines in Mexico with varying capacities and design standards using an analytical approach [E1.17-22], showing that wind turbines have a low fragility for earthquakes. However, this may be attributed to the assumption about the soil characteristics in the model; stiff-soil conditions were assumed close to the source, whereas soft-soil conditions may lead to higher fragilities. For nuclear power plants (NPPs), MI (2019) present adapted fragility curves [E1.13-14], representing a 'complete' damage state, for NPPs with a fixed base (i.e., non-seismic design) and an isolator (i.e., seismic design). The latter is obtained from Ahmad et al. (2015), which used an analytical approach to assess the vulnerability of a NPP reactor building with a height of 65.8 m with a specific structural design.

FEMA (2020) developed fragility curves for substations with the probability as a function of PGA [E2.1-6]. They defined four damage states, for which fragility curves are developed similarly to power plants. For a global application, MI (2019) adjusted the high-voltage unanchored substations vulnerability curve provided by FEMA (2020) based on expert-opinion to account for a higher expected vulnerability and a lower quality [E2.7-8]. Omidvar et al. (2017) provide a set of fragility curves for low-voltage unanchored substations with PGA as intensity measure [E2.9] and the similar damage states as maintained by FEMA (2020). López et al. (2009) applied an analytical approach to develop fragility curves for substations using spectral pseudo acceleration as the intensity measure [E2.10-11]. The substation is representative for lattice frame substations in Mexico with a 400 kV double switch.

Zheng et al. (2017) perform an explicit dynamic analysis to calculate the probability of seismic collapse of a typical high-rise power transmission tower in China [E3.1]. Hereby, various factors are taken into account, such as member failure rule, the amount of dead weight, the tower height and different ground motion inputs. Also, three failure mechanisms are explicitly considered, namely strength failure, ultimate strain failure, and compression member buckling and softening failure. Long et al. (2018) developed a fragility curve to represent the collapse probability of steel power towers with a height of 21 m subject to unidirectional earthquake ground motions [E3.2]. Sadeghi et al. (2012) apply a non-linear dynamic approach to develop a

fragility curve for tubular steel poles that are characterized by a height of 19.5 m and are used for 63 kV transmission lines [E4.5].

FEMA (2020) developed fragility curves for T&D circuits that consist of either anchored or unanchored components, again using PGA as hazard intensity measure. They defined four damage states, for which fragility curves are developed similarly
to power plants. These vulnerability curves are also applicable to poles [E4.1-2], wires, other in-line equipment and utility-owned equipment at customer sites and can be applied to underground [E5.1-2] and elevated [E6.1-2] infrastructure (FEMA, 2020). For a global application, MI (2019) adjusted the unanchored distribution vulnerability curve provided by FEMA (2020) to account for a higher expected vulnerability and a lower quality [E4.3-4, 5.3-4 and 6.3-4].

### 3.2.2 Transportation

Maruyama et al. (2010) provide vulnerability curves for roads that express the number of damage incidents per km against PGV using a compiled database consisting of damage data for three earthquakes [E7.15]. Argyroudis et al. (2018) apply an analytical approach to develop fragility curves for three damage states ('minor', 'moderate', and 'extensive/complete') for the representation of highways [E7.1-5] and railways [E8.11-15] on an embankment. They analysed the joint effect of flooding and earthquake by using a range of inundation depths as a precondition for their model. Argyroudis and Kaynia (2015) provide
fragility curves for road [E7.6-15] and railways [E8.16-24] on an embankment and in cuts, thereby considering two soil conditions (i.e., soil type C and D following the Eurocode 8) and a range of embankment heights (i.e., 2, 4 and 6 m). Shinoda et al. (2022) present analytically derived fragility curves for railway embankments conform to Japanese design standards, using typical design parameters for checking the stability of embankments. The set limit state corresponds to a seismic displacement of 50 cm in the crest of the embankment, meaning that substantial time is needed to repair the damage. The curves from this
Japanese study address the presence of a primary reinforcement, including its tensile strength, and the friction angle of the backfill soil [E8.25-59].

FEMA (2020) provides curves for the estimation of earthquake damage to infrastructure types categorized under the railway transportation system. According to FEMA (2020), railway facilities (e.g. maintenance, fuel, and dispatch facilities), bridges
and tunnels are vulnerable to both ground shaking and ground failure, while tracks and roadbeds are particularly vulnerable to ground failure alone (see Section 3.4 for ground failure curves). Fragility curves for fuel and dispatch facilities are developed with respect to seismic design (unanchored vs. anchored) and whether the facility has a backup power system, with PGA as intensity measure [E8.1-8]. The curves of these facilities are based on the potential damage that may occur to their subcomponents, such as the pump building, electric power, and tanks, using Boolean expressions. Furthermore, fragility curves
are presented for low voltage direct current (DC) power substations that convert electrical power specifically for light rails [E8.9-10], and are developed using a similar methodology (FEMA, 2020).

Following the categorization of FEMA (2020), airports consist of the infrastructure types of runways, control towers, fuel facilities, maintenance and hangar facilities, and parking structures. Potential damages to runways are described by ground failure (see Section 3.4) as ground shaking is not a large source for damages to these structures. Fragility curves for airport fuel facilities are assumed to be similar to railway fuel facilities [E9.1-4]. For the remaining facilities, the standard building fragility curves for a selection of building categories provided by FEMA (2020) can be applied, which are not included in our database.

Vafaei and Alih (2018) selected three in-service Air Traffic Control (ATC) towers with different heights, ranging between 24 and 52 m, but similar structural systems, for an analytical seismic fragility assessment. Three damage states are defined. The first damage state 'immediate occupancy' indicates that structures require little or no repair after an event. The second damage state 'life safety' indicates significant damage to the structures, but that these structures still provide a reasonable safety margin against collapse. The last damage state 'collapse prevention' indicates that the structures continue to support gravity loads, but that there is no safety margin against collapse. The fragility curves show that the higher towers are significantly more vulnerable to earthquakes and that towers are more susceptible to the low category of PGA/PGV ratios [E9.5-13].

### 3.2.3 Telecommunication

A set of fragility curves are developed for the seismic structural performance of monopole towers with a height of 24 m in Iran (Sadeghi et al., 2010). Three limit states are defined (i.e., 'low', 'medium', 'severe') and used to derive fragility curves analytically [E11.1]. Fragility curves for central offices and broadcasting stations [E12.1-2] are developed with respect to their seismic design (FEMA, 2020). These curves are based on the probabilistic combination of curves for components of the communication facility (e.g., power backup system, switching equipment and building) using Boolean expressions to describe the relationship of these components to the communication facility.

### 3.2.4 Water

PGA-related fragility curves are developed by FEMA (2020) for storage tanks, accounting for construction material and elevation level. Two sets of fragility curves are provided for on-ground steel storage tanks with respect to their seismic design [E13.3-4], and one for above-ground steel storage tanks [E13.5]. One set of fragility curves is provided for on-ground wooden storage tanks [E13.6] and another two for on-ground concrete tanks [E13.1-2]. MI (2019) expect a higher vulnerability of elevated and ground level unanchored storage tanks compared to the FEMA (2020) fragility curves, and adjusted them accordingly based on expert-opinion [E13.7-10].

Eidinger et al. (2001) developed fragility curves for storage tanks with varying fill levels and seismic designs based on a compounded damage database. Four sets of fragility curves are derived for fill levels of <50%, ≥50%, ≥60%, and ≥90%, showing that storage tanks with low fill levels (<50%) have a higher vulnerability compared to ones with a high fill level

[E13.11-17]. Two sets of fragility curves are presented for storage tanks with a fill level of 50% and two seismic designs (i.e., anchored and unanchored), showing that the median PGA value to reach various damage states is 3-4 times higher for anchored storage tanks than for unanchored tanks. We refer to Eidinger et al. (2001) for a range of analytical fragility curves for specific damage states (e.g., tank slides break inlet line). O'Rourke and So (2000) also developed fragility curves based on empirical data on the seismic structural performance during nine earthquake events. They applied the damage state descriptions for

storage tanks from the HAZUS methodology to develop: (1) a general set of fragility curves; and (2) curves that take into account physical characteristics (i.e., diameter/height ratio and the relative amount of liquid stored) of on-ground steel liquid storage tanks [E13.18-22]. In comparison with the FEMA (2020) curves, the O'Rourke and So (2000) curves envision a higher structural performance (i.e., lower vulnerability). Berahman and Behnamfar (2007) use a Bayesian statistical technique to assess the fragility of unanchored on-grade steel storage tanks with a fill level above 50% and without attributes [E13.23-24].

A more recent study used a compiled database with 5,829 above-ground steel liquid storage tanks from 24 seismic events to develop fragility curves (D'Amico and Buratti, 2019), showing that a tank has a lower seismic performance if it is slender, unanchored and has a low filling level [E13.25-31].

FEMA (2020) developed six sets of fragility curves for WTPs, with each set consisting of a 'Slight', 'Moderate', 'Extensive',

and 'Complete' damage state. Two sets are devoted to each of the three WTP categories, which are based on capacity and are developed for a seismic and a non-seismic design. These fragility curves are described using PGA as hazard intensity measure and are based on the probabilistic combination of damage curves for components (e.g., sedimentation tanks, chlorination tanks, and electric power) of the WTP through the use of Boolean expressions [E14.1-6]. One set of PGA-related fragility curves for wells is presented by FEMA (2020), which assumes that the equipment is anchored [E15.1]. The components power backup,

well pump, building and electrical equipment are applied to develop these fragility curves by using Boolean expressions.

Wave propagation damage to buried pipelines may occur over wide geographical areas, and therefore O'Rourke and Ayala (1993) developed a curve based on observed pipeline damages due to earthquakes in the US and Mexico [E16.1]. An empirical relation is established between the PGV and a repair rate that expresses repairs needed for each km of brittle pipeline. Ductile

pipelines, which are more flexible, are expected to have a lower vulnerability compared to brittle pipelines (O'Rourke and Ayala, 1993). FEMA (2020) adapted this curve to represent the fragility of brittle pipelines, and a 30% lower fragility is assumed for ductile pipelines (including pipelines made of steel, ductile iron and PVC) [E16.2-3]. Eidinger et al. (2001) use empirical data from 18 earthquakes to develop vulnerability curves expressed as the repair rate as the function of PGV [E16.4-22].

Piccinelli and Krausmann (2013) compiled 26 empirical studies relating pipeline damage to ground shaking effects for the period 1975-2013. Kakderi and Argyroudis (2014) provide an overview of literature containing functions expressed as repair rate (repairs per km) and breaks per pipe length for the similar period. In addition to the aforementioned reviews, Shih and

Chang (2006) present empirical vulnerability curves for PVC water pipes in China for both PGA and PGV, using data derived from the 1999 Chi-Chi Taiwan earthquake that caused widespread damages to the underground pipeline infrastructure [E16.23-24]. Using observations from the 1985 Michoacan earthquake, Pineda-Porras and Ordaz (2012) developed an empirical fragility curve describing repair rate and the composite parameter $PGV^2/PGA$ as a metric for ground motion [E16.25]. Yoon et al. (2018) developed fragility curves for cast and steel pipes that have been buried for 20 and 30 years by explicitly considering the impact of deterioration. Compared to steel pipes, cast iron pipes deteriorate rapidly and have a high fragility [E16.26-31]. Sadashiva et al. (2021) derived vulnerability curves for buried pipelines, which are categorized by pipe size and material type, based on damage records of the water supply network due to the 2010-2011 Canterbury earthquake sequence that was accompanied with widespread and severe liquefaction [E16.32-39].

FEMA (2020) developed four sets of fragility curves, with each set consisting of a 'slight', 'moderate', 'extensive', and 'complete' damage state, for pumping plants with respect to their capacity [E17.1-4]. Half of the fragility curves represent a seismic design, while the other half represent a non-seismic design. These fragility curves are described by PGA as hazard intensity measure and are based on the probabilistic combination of damage curves for components (e.g., power backup system, pumps, and other electrical equipment) of the pumping plant using Boolean expressions to describe the relationship of these components to the pumping plant. Finally, to our knowledge, no damage curves exist for control vaults and stations.

### 3.2.5 Waste

Fragility curves were developed by FEMA (2020) for WWTPs with respect to seismic design and capacity. Half of the curves present a design with anchored components in the WWTP, and the other half present a WWTP without anchored components. The curves represent higher vulnerability with lower capacities, and PGA is applied as hazard intensity measure [E18.1-6]. Boolean expressions are applied probabilistically to describe the relationship between WWTP components (e.g., sedimentation tanks, chlorination tanks, and electric power). Liu et al. (2015) developed empirically derived vulnerability curves for sewer gravity and pressure pipes using damage records collected after the 2010-2011 Canterbury earthquake sequence [E19.3-43]. Nagata et al. (2011) developed fragility curves for sewerage pipes based on seismic damage data for a number of earthquakes that occurred in the period of 2004-2008 in Japan [E19.44-46]. They define the damage ratio as the proportion of the total length of damaged pipes to the total length of sewerage pipes, and use this to describe the relationship with the Maximum Ground Velocity (MGV). The fragility curve for pipelines situated in areas with a liquefaction potential shows higher damage ratios versus non-liquefaction pipelines. The FEMA (2020) curves for brittle and ductile pipelines (See Section 3.2.4 Water) can also be applied to sewers and interceptors [E19.1-2]. For lift stations, the vulnerability curves are similar to those for pumping plants presented in Section 3.2.4 [E20.1-4]. To our knowledge, no curves exist for control vaults and stations.

### 3.2.6 Health & Education

Giordano et al. (2021b) developed empirical fragility curves for main structural school typology buildings in Nepal. Through the World Bank's Global Program for Safer Schools, an empirical database was developed following the 2015 Nepal earthquake, which contains post-earthquake data for approximately 18,000 Nepalese school buildings. For four building classes (i.e., masonry, reinforced steel frame, steel frame and timber frame) fragility curves were estimated for damage states "slight", "moderate", "extensive" and "collapse" [E21.1-4]. Another set of fragility curves for Nepalese school buildings is

developed by Giordano et al. (2021a). In their study, they present analytical fragility curves for three types of unreinforced masonry school buildings: rubber stone mud (URM-SM), brick-mud (URM-BM), and brick-cement masonry (URM-BC) buildings. The number of stories is also considered [E21.5-10].

Hancilar et al. (2014) developed fragility curves for typical public school buildings in Turkey (i.e., a four-story reinforced

concrete shear wall building with moment resisting frames) through a probabilistic analytical approach. We include their curves for PGA, PGV and elastic spectral displacement as hazard intensity measures for ground motion [E21.11-13], and refer to Hancilar et al. (2014) for the sets of fragility curves considering adjusted parameters. D'Ayala et al. (2020) provide non-retrofitted and retrofitted fragility curves for two-story reinforced concrete (RC) frame and three-story school buildings that are typically used for primary and secondary education in the Philippines [E21.14-17]. Samadian et al. (2019) provide fragility

curves for concrete and RC school buildings in Iran [E21.18-19] and Baballëku and Pojani (2008) provide fragility curves for RC school buildings in Albania following an analytical approach [E21.20]. For a hospital in San Francisco, Ranjbar and Naderpour (2020) developed fragility and vulnerability curves using earthquake records based on the distance from the fault. They show that the hospital has a higher fragility if exposed to near-field earthquakes (i.e., less than 10 km) compared to far-field earthquakes (i.e., equal or greater than 10 km) [E21.21-22].


FEMA (2020) provides damage curves for general building stock, which can also be applied to hospitals (with varying capacities), medical clinics, and educational facilities (i.e., schools and colleges/universities). Fragility curves ('None', 'Slight', 'Moderate', 'Extensive', and 'Complete') are developed for a range of building categories, further specified by building characteristics (e.g., height) and levels of seismic design (FEMA, 2020). For building stock in a European context,

Milutinovic and Trendafiloski (2003) provide fragility and vulnerability curves for 65 building classes, using the European Macroseismic Scale (EMS-98) and spectral displacement as intensity measure.

### 3.3 Windstorms

### 3.3.1 Energy

We did not find curves for power plants in general, whereas we did find curves for different power plant types. Watson and

Etemadi (2020) provided fragility curves of hurricane wind conditions for coal, gas and nuclear plants, solar panels, and wind

turbines by adapting existing curves, such as the HAZUS building damage curves [W1.10-14]. For a risk assessment of wind turbines in Mexico, Jaimes et al. (2020) developed fragility and vulnerability curves for wind turbines with hub heights of 40, 80 and 100 m [W1.1-3]. Martín del Campo et al. (2021) extended the previous study by analysing the effect of passive damping systems, which can reduce the fragility under wind attacks by approximately 80% [W1.4-9]. López et al. (2009) developed
fragility curves for substations considering two design types, one for wind speeds of 200 km/h and another for wind speeds of 300 km/h [W2.1-2]. Watson and Etemadi (2020) provide fragility curves for substations that are based on internal data of FEMA [W2.3-7], considering three damage states as a function of the peak wind speed. Also, the following terrain types are considered: open, light suburban, suburban, light urban and urban. Substations situated in open areas have a higher vulnerability compared to substations located in areas with a higher building density.


Raj et al. (2021) developed a set of fragility curves for lattice transmission towers based on damage records in India during the 2019 cyclone Fani. Two limit states were defined: 'partial' and 'collapse' for high-voltage towers (132-220 kV) characterized by heights between 21-50 m [W3.15]. López et al. (2009) developed fragility curves for small and tall lattice transmission towers considering two design types (i.e., wind speeds of 120 km/h and 160 km/h). The structural model was tested similarly
to substations, but now with consideration for the tension in cables due to wind loading [W3.1-4]. Hur and Shafieezadeh (2019) present an analytically derived fragility curve for a transmission tower representative for lattice type towers in the coastal areas of southeast US [W3.44]. Fu et al. (2019) developed a set of fragility curves for a transmission tower (500 kV) under wind loading [W3.45], and fragility curves considering the wind direction and the orientation of the transmission tower [3.47-50].

Reinoso et al. (2020) assessed the vulnerability of transmission towers by explicitly considering the coupling of the tower with overhead lines. The failure mechanism is based on the capacity considering the collapse probability of the tower and intermediate levels of damage. This resulted in a range of vulnerability curves for transmission towers with various design wind speeds on an urban terrain [W3.5-14]. Also, Cai et al. (2019) developed fragility curves for a range of wind attack angles and horizontal spans for typical towers in China by explicitly considering the tower-line coupling [W3.26-43]. Xue et al. (2020)
also evaluate the fragility of a transmission tower-line system instead of a stand-alone transmission tower. Fragility curves are given for stand-alone towers and for the transmission towers as coupled systems, for five wind attack angles [W16-25]. Quanta Technology (QT, 2009) provide fragility curves for regular and hardened transmission structures with a wind loading standard of 169 and 209 km/h, respectively, drawing upon historical records over a 10-year period [W3.51-52]. Panteli and Mancarella (2017) present an analytically-derived fragility curve for transmission towers in the UK [W3.53].


González de Paz et al. (2017) developed fragility curves for wood poles in Argentina, employing five distinct models [W4.75-79]. QT (2008) provide fragility curves for poles in the US based on historical records from the private sector [W4.80]. The following studies concern the fragility of typical Southern pine utility poles in the US. Bjarnadottir et al. (2013) present fragility curves for four design classes with varying strength and load factors and for four pole ages [W4.1-16]. They considered the

load of components, such as conductors and wires, to compute the design load of the poles, and they included the deterioration effect in their modelling. Shafieezadeh et al. (2014) developed fragility curves for two common classes of Southern pine poles. The damage state of interest is the breakage of the utility poles, and a model for the representation of deterioration of the wood is included. The fragility curves are the result of a Monte Carlo simulation that compares realizations of the demand and capacity across a wide range of wind velocities [W4.17-26]. Han et al. (2014) combined a structural reliability model for utility poles with damage records from hurricanes Katrina, Dennis and Ivan in the Central Gulf coastal region through Bayesian updating [W4.27-28]. The outcome is a fragility curve for Southern Pine utility poles and for two horizontal spans. Salman and Li (2016) considered the fragility of Southern pine and steel poles in the US taking into account deterioration over time due to wood decay and steel corrosion [W4.29-36]. Also, they explicitly included the load of the wires in the modelling of flexural failure. Yuan et al. (2018) also developed fragility curves for Southern pine poles (class 2 and 3 and different pole ages) in the US [W4.37-44]. They used a finite element model of a three-span pole-wire to perform a non-linear finite element analysis, considering the load on the pole and wire as well as the age-deterioration effect. For the latter, the age-deterioration model of Shafieezadeh et al. (2014) is adapted. Fragility curves for wooden poles under extreme wind conditions in the US were developed using a Monte Carlo simulation by Lee and Ham (2021), thereby also considering the effect of strength degradation over time due to wood decay. Fragility curves are given for one design type, two pole ages and four angles of leaning [W4.45-52]. Whereas the previous models are often generic with specific assumptions about (1) the configuration and properties of the structure, and (2) the wind direction that is perpendicular to the conductors representing a worst-case scenario (e.g., Salman and Li, 2016; Shafieezadeh et al., 2014), Darestani and Shafieezadeh (2019) point out the necessity of the development of multi-dimensional fragility curves that account for multiple parameters, and provide them for a range of Southern Yellow Pine wood pole design types [W4.53-66]. Teoh et al. (2019) developed a probabilistic performance assessment for Southern pine poles exposed to winter storms, taking into account both ice formation and wind speed. Using a finite element model composed of three poles in combination with generated wind speeds, a finite element analysis was performed. Fragility curves were developed for a range of design types, pole ages and mitigation strategies [W4.67-74].

Dunn et al. (2018) used a database containing information on faults to the electrical distribution system in the UK, including data on faults to overhead lines due to windstorms. Fragility curves are constructed for 11-132 kV overhead lines, whereby the fragility is presented as the mean of the number of faults per km against the wind speed. The probability can be obtained by dividing this by the average length of overhead lines between poles [W6.1]. Panteli et al. (2017) present fragility curves for overhead transmission lines in the UK [W6.2], and QT (2008) for overhead lines in the US based on historical records from the private sector [W6.3].

**3.3.2 Transportation**

Zhu et al. (2022) assessed the vulnerability of roads to tropical cyclones and their joint effect of precipitation and wind speed by using damage records from events in Hainan Province, China. These records include damage observations to various

structures such as protection components of a road, pavement and subgrade. In our database, we include the physical damage probability curve that applies the maximum wind speed at 10 m above ground-level as the intensity measure [W7.1] and refer to Zhu et al. (2022) for the multi-variate curve for the concurrent compound hazard intensities. According to MI (2019), on-grade roads are not vulnerable to direct wind damage [W7.2]. We did not find any vulnerability curves for railways and airports.

### 3.3.3 Telecommunication

Gao and Wang (2018) performed a nonlinear dynamic analysis by applying the alternative load path method. A finite element model was developed for two standardized types of lattice towers commonly built in China, namely a 50 m high tripole (i.e., three supporting legs) and angle (i.e., four supporting legs) tower. They examined for a range of wind directions and leg member failures in order to determine the probability of structural collapse and found a higher vulnerability for tripole towers compared to angle towers [W10.1-2]. Bilionis and Vamvatsikos (2019) focused on a standardized type of lattice tower used in Greece that is designed according to European Standards for structures built within 10 km of the coastline [W10.3]. Tian et al. (2020) developed curves for the probability of structural collapse for an angle latticed tower using a dynamic explicit method [W10.4-9], specifically considering member buckling as a failure mechanism. Their results demonstrate that the wind attack angle has a significant impact on the collapse fragility curve, and that main members were the governing reason for the progressive collapse of the structure, similar to Gao and Wang (2018).

### 3.3.4 Water

Ground level tanks are generally not affected by wind loading unless the wind forces are exceptional, whereas elevated water tanks have a higher probability to get damaged (MI, 2019). A range of variables influence the level of vulnerability such as the tank filling level (Olivar et al., 2020) and roof configuration (Virella et al., 2006). However, we did not find vulnerability or fragility curves for water tanks. WTPs are low structures and are not vulnerable to damage from wind loading [W14.1]. Buried water pipelines are not adversely affected by windstorms [W16.1] (MI, 2019). Furthermore, we did not find any curves for water wells, pumping plants, water control vaults or stations.

### 3.3.5 Waste

WWTPs are low structures that are not vulnerable to wind damage [W18.1], and also buried water pipelines are not adversely affected by windstorms [W19.1] (MI, 2019). Furthermore, we did not find any curves for the sewer and interceptor network, lifts, waste control vaults and stations.

### 3.3.6 Health & Education

Acosta et al. (2018) developed vulnerability curves for a range of design types of 1-storey school buildings in the Philippines, specifically focusing on damage to the building envelope (i.e., roof fastener, ceiling board and windows), using field surveys [W21.1-5]. Acosta (2022) focused on 1-storey school buildings with wooden roof structures [W21.6-10]. The fragility curves

show that the roof-to-column connection has a low impact on the vulnerability of schools, while the environment has a significant impact with structures in open areas having a higher vulnerability compared to urban and suburban areas. Also, a vulnerability curve is constructed based on the modelled fragility curves and compared to field survey data of Typhoon Nina. FEMA (2021a) provides fragility curves for the general building stock in their technical manual for hurricanes. Elementary schools, high schools and hospitals are explicitly modelled using a component-based approach, whereby the first two are characterized by low-rise structures and the latter can be both low- and high-rise in nature.

## 3.4 Landslides

### 3.4.1 Energy

Permanent ground deformation (PGD) is a measure to express ground failure that is caused by liquefaction, landslides and surface fault rupture (FEMA, 2020). We therefore include curves with PGD as hazard intensity measure as they also express the vulnerability to landslides. FEMA (2020) assumes that the curves due to ground failure for power plants [L1.1] and substations [L2.1] to be similar to those described for potable water system facilities (Section 3.4.4.). Glade (2003) assume that power lines in North-western Iceland will be completely destroyed by debris flows [L6.1] and rock falls [L6.2] of low, medium and high intensity.

### 3.4.2 Transportation

Roads are significantly affected by ground failure, while bridges and tunnels are vulnerable to both ground shaking and ground failure (FEMA, 2020). FEMA (2020) provides a set of fragility curves for major roads (i.e., roads with four lanes or more, and parkways) [L7.1] and urban roads (i.e., roads with two lanes) [L7.2] using PGD as intensity measure. Glade and von Davertzhofen (2003, as cited in Glade, 2003) pragmatically assume that motorways and country roads in the way of a landslide are completely destroyed in Germany [L7.3-4]. For roads in Australia, Michael-Leiba et al. (2000, as cited in Glade, 2003) also work with such fixed damage values: 0.3 for landslides on hill slopes, 1 for roads at the origin of a debris flow, and 0.3 at the deposition location of a debris flow [L7.5-7]. Likewise, Remondo et al. (2008) provide fixed damage values for shallow landslides in Spain, for four different road types [L7.8-11] and for railway [L8.4]. Zêzere et al. (2008) provide fixed damage values for roads in Portugal: 0.6 for rainfall-triggered shallow translational slides, and 1 for translational and rotational slides [L7.12-23].

Glade (2003) provide damage values for two landslide types (i.e., debris flow and rock flow) by magnitude category (i.e., low, medium, high) [L7.24-25]. For debris flow the vulnerability value can be up to 0.6, while this is 0.4 for rock falls. Leone et al. (1996, as cited in Glade, 2003) present damage values for four categories of damage intensity and associated type of damage [L7.26]. For landslides at cut slopes, i.e. where the mountain was excavated to make place for the road or rail, Jaiswal et al. (2010) also apply fixed damage values in three magnitude classes, for both asphalt roads [L7.27] and railroads [L8.5].

Likewise, Jaiswal et al. (2011) use three magnitude classes for rapid debris slides in India, for which they provide minimum, average and maximum damage values for roads [L7.28-29] and railroads [L8.6-7]. Based on landslide records for an Himalayan road corridor road in India, Nayak (2010) provide damage values for debris and rockfall landslides with different magnitudes [L7.30-31].

For slow-moving landslides in Italy, Galli and Guzzetti (2007) express the vulnerability of major and secondary roads as a function of landslide area that serves as proxy of hazard severity [L7.32-33]. Empirical fragility and vulnerability curves [L7.34] for road networks exposed to slow-moving landslides in Italy are developed by Ferlisi et al. (2021). They also present time-dependent vulnerability curves to account for an increasing vulnerability of roads due to an increase of cumulative displacements of interacting slow-moving landslide bodies over time [L7.35-37]. Winter et al. (2014) develop fragility curves for high-speed and low-speed roads exposed to debris flows by expressing the probability of three damage states as a function of the debris flow volume based on expert-judgement [L7.38-39]. Nieto et al. (2021) express hazard severity in 'debris flow height', for two road types and for variable embankment heights [L7.40-45].

Railway tracks are significantly affected by ground failure, while other elements of the railway system, such as bridges and tunnels, are vulnerable to both ground shaking and ground failure (FEMA, 2020). FEMA (2020) provides a set of landslide fragility curves for railway tracks [L8.1], which are assumed to be similar to those of major roads, and fuel facilities with buried tanks [L8.2]. The curves for other elements of the railway system (i.e., stations, maintenance- and dispatch facilities) are similar as those described for buildings (see FEMA (2020) for further reference). Argyroudis and Kaynia (2014) describe three damage states in terms of permanent ground displacement for railway [L8.3] based on a review of the literature that are used to develop fragility curves. Martinović et al. (2016) present a fragility curve for rainfall-triggered shallow landslides, which expresses failure probability as a function of rainfall duration, for different slope angle values [L8.8-11]. Zhu et al. (2023) provide vulnerability curves for Chinese railway in context of rainfall-induced hazards including landslides. Curves are developed at national [L8.12] and subregional scale [L8.13-18] using historical damage records, precipitation data and infrastructure market values.

FEMA (2020) provides a set of fragility curves for paved runways described by ground failure [L9.1]; ground shaking is not a large source for damages to these structures. Curves for airport fuel facilities [L9.2] are similar to railway fuel facilities. For other airport facilities, the standard building fragility curves for a selection of building categories provided by FEMA (2020) can be applied, which are not included in our database.

### 3.4.3 Telecommunication

FEMA (2020) assumes that curves due to ground failure for communication facilities (i.e., central offices and broadcasting stations) [L12.1] are similar to those described for potable water system facilities (Section 3.4.4.).

### 3.4.4 Water

For water storage tanks [L13.1], WTPs [L14.1], wells [L15.1] and pumping plants [L17.1], FEMA (2020) assume that there is a 50% chance of complete damage for 0.25 m of PGD. The other damage states are assumed to be similar to those described for buildings (see FEMA (2020) for further reference). For buried concrete tanks, a set of fragility curves is provided using PGD as the intensity measure [L13.12]. Eidinger et al. (2001) provide fragility curves for water storage tanks based on expert-judgement using PGD as intensity measure [13.2-13.11]. Ground failure generally causes breakage to a pipe while seismic wave propagation causes leaks due to, for example, joint pull-out and crushing at the bell (e.g., Kakderi and Argyroudis, 2014; FEMA, 2013; MI, 2019). Eidinger (1984) and Eidinger et al. (2001) provide empirical curves for estimating pipe repairs due to PGD for a range of pipe materials and joinery types [L16.1-16.11 and L16.12-16.24]. A damage model for buried brittle and ductile pipelines due to ground failure is presented in FEMA (2020) where the repair rate is a function of PGD and the probability of an event.

### 3.4.5 Waste

FEMA (2020) assumes that the curves due to ground failure for WWTPs [L18.1] and lift stations [L20.1] are similar to those described for potable water system facilities, and that the damage models proposed for buried pipelines in potable water systems can be applied to sewers and interceptors in the waste system [L19.1-24] (see Section 3.4.4.). Furthermore, we did not find any curves for the waste control vaults and stations.

### 3.4.6 Health & Education

Konovalov et al. (2019) defined four categories by thickness of sliding mass, estimated landslide volume and magnitude class to develop a simplified vulnerability model for schools in Russia. For each category two sets of damage values are provided [L21.1-2]. Furthermore, FEMA (2020) assume that the ground failure damage curves developed for the general building stock can be applied to health and education facilities.

## 4 Comparison of vulnerability data across hazard types and CI types

In this paper we synthesised state-of-the-art knowledge about fragility and vulnerability curves for various hazards and CI types. The main contribution of this paper is to extract all this knowledge into a novel database that is useful for the wider research community. In this section, we identify cross-hazard and cross-infrastructure data issues we encountered in our work and discuss opportunities for learning.

## 4.1 Coverage of data across hazards and CI types

Our database contains 803 sets of fragility and vulnerability curves, with the curves almost evenly distributed over both curve
types (54% and 46%, respectively). If the curves for the damage states within a fragility set and curves for the uncertainty
bandwidths are accounted for separately, the database counts over 1,510 unique curves. An overview of the distribution of
curve sets in our database per hazard type and CI system, as well as the distribution over time is provided in Fig. 4. In Fig. 4a
is shown that curves are predominantly focused on energy (30.8%), followed by transportation (25.8%), and water (20.9%),
whereas the other CI systems have a substantially smaller variety of available curves, with the telecommunication system
being largely underrepresented (2.2%). Telecom assets are vulnerable to natural hazards, and disruption of these assets may
impede disaster recovery efforts that rely on a readily available communication (Sandhu and Raja, 2019; Marshall et al., 2023).
While all CI systems require more research, we emphasize the need to put additional efforts in telecom. Furthermore, health
and education facilities are represented by a small share of curves (6.6%) in our database. However, curves are available for
the general building stock such as provided by FEMA (2013; 2020; 2021a) and Milutinovic and Trendafiloski (2003).


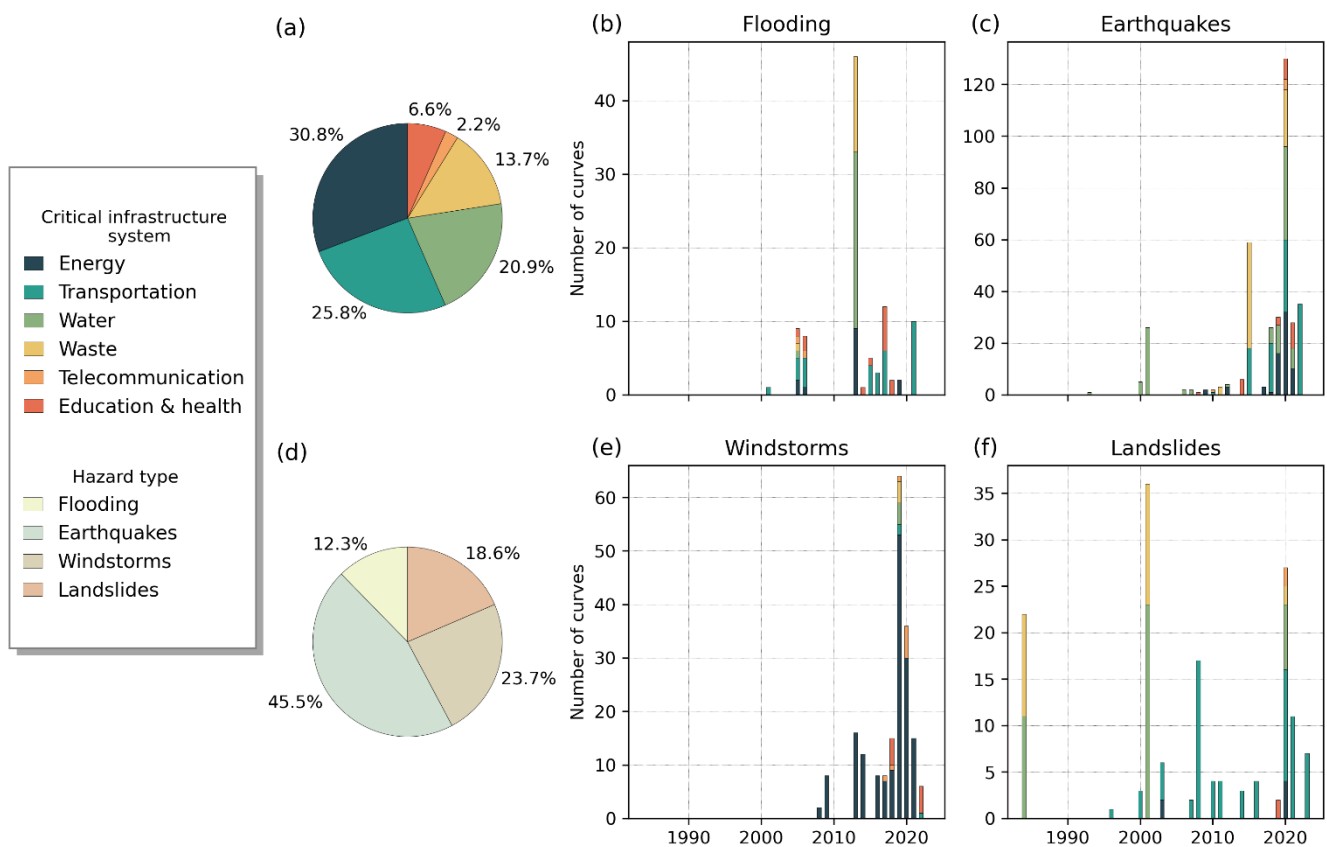

**Figure 4: Panel A presents the relative share of curves in our database per CI system, whereas Panel D presents the relative share of curves per hazard. Panel B, C, E and F present the distribution of curves over time (1984-2023), subdivided by CI system. In case of fragility data, we count the set of fragility curves rather than each curve within a set.**


Vulnerability research in the context of CI increasingly received attention from 2000 onwards (Fig. 4b-c,e-f); 80.0% of the curve sets were published in the period 2010-2022. Figure 4d clearly highlights that most curves for CI over the past years are focused on earthquakes (45.5%), with the majority of the curves developed for the transportation (27.7%) and water system (26.8%). Notably, 84.2% of the wind curves represents the energy system, whereas the representation of the other CI systems

is substantially lower (Fig. 4e). This lower number of curves can be partly attributed to lower levels of susceptibility to gust speeds for several infrastructure assets (e.g. water treatment plants or sewage systems). However, other infrastructure assets require more research. For example, we would have expected to find wind curves for airports, where damages to hangers and airplanes have been observed in the past (Özdemir et al., 2018). Also, infrastructure may be susceptible to secondary hazards associated with wind (QT, 2008): tree fall and flying debris may lead to structural damage to, for example, railways (Palin et

al., 2021). Furthermore, landslide curves are predominantly focused on transportation (48.3%) and water (27.5%), while the share of flood curves across the CI systems is more balanced.

## 4.2 Geographical coverage

The geographical application of the collected sets of fragility and vulnerability curves is shown in Fig. 5 using a percentile distribution. The database predominantly contains curves that are presented as country-specific, but also curves with a wider

geographical application. We found 86 curve sets for application at the global scale, which were retrieved from MI (2019), Huizinga et al. (2017), and Winter et al. (2014), and from ten other sources for which we assume a global application, such as the curves provided by Nieto (2021) for mountainous areas. Curve sets for regional applications were also provided by multiple sources, such as the flood curve for trans-boundary rhine countries (ICPR, 2001). In general, the database has an above median coverage for Europe, Asia and North America, whereas South America, Central America, and, especially, Africa are

underrepresented. We found that the coverage for the US is the highest: a total number of 428 curve sets are applicable to the US, followed by Japan (164), and Mexico (154).

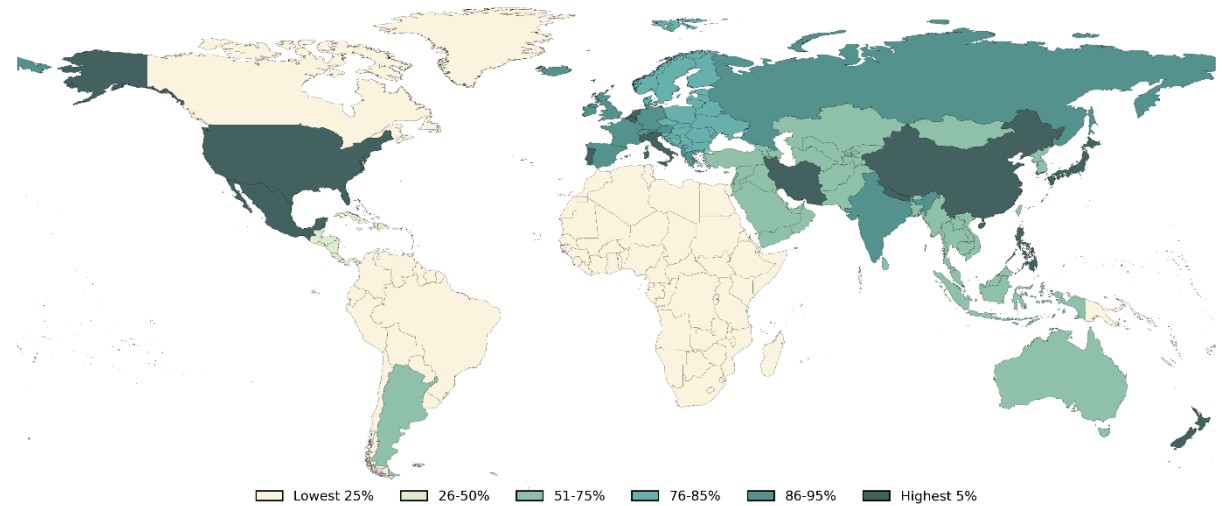

**Figure 5: Geographical distribution of the sets of fragility and vulnerability curves collected within the database. In case of fragility data, we count the set of fragility curves rather than each curve within a set. Uncertainty bandwidths are not accounted for separately. Curves without a specified geographical application in the original source are assumed to have a global application.**

### 4.3 Characteristics of curves across hazards

We found that flood vulnerability to assets is typically quantified in terms of vulnerability curves. FEMA (2013) is an exhaustive source for such information, and contributes with 46.5% to the flood curve sets in our database. Earthquake and wind vulnerabilities are typically quantified through fragility curves for one or multiple damage states. On the contrary, the vulnerability of landslides is generally quantified in terms of a fixed damage value if a (specific type of) landslide (of a certain magnitude class) occurs (e.g., total destruction if exposed to landslide) without explicitly considering a hazard intensity measure. A landslide is a complex phenomenon that can be triggered due to different hazards, such as earthquakes, volcanic eruptions, windstorms and rainfall, and results in the creeping, toppling, sliding or flowing of material such as rock, debris and soil, down a slope. Due to this complexity, we found a wide variety of hazard intensity measures, such as rainfall intensity, debris flow height, and the volume.

We encounter a range of ground shaking hazard intensity measures for earthquakes such as PGA, PGV and elastic spectral displacement. Conversely, flooding predominantly relies on a single intensity measure, inundation, although there is a rare instance where WIM is used. However, other intensity measures such as flow velocity (Kreibich et al., 2009; Koks et al., 2022) and salinity (Glas et al., 2017) also play an important role to infrastructure damage. The focus on depth-damage curves seems mainly driven by the pragmatic consideration that inundation depth is the most common and easy to calculate metric for flood hazard data, despite evidence that flow velocity is a better indicator for structural damage to, for example, bridges (e.g., Koks et al., 2022). Kreibich et al. (2009) even argue that: "Forecasts of structural damage to road infrastructure should be based on flow velocity alone.". The selection of the correct hazard intensity measure to representatively describe the vulnerability of an

asset is crucial. For example, within the earthquake domain, Pineda-Porras and Ordaz (2012) even introduced a new metric to better depict pipeline damage in specific local soil conditions. Moreover, recent studies have also begun to assess the vulnerability of CI due to the joint effect of multiple hazards (e.g., Argyroudis et al., 2018; Teoh et al., 2019; Zhu et al., 2022), aligning with the growing field of multi-hazard research aimed at elucidating the interactions of hazards (e.g., Gill and Malamud, 2014; Lee et al., 2024).

Vulnerability quantification methods have historically been more advanced in the earthquake and wind risk modelling, where the curves are dominantly analytically derived using methods that have a strong focus on object-based physical attributes (De Ruiter et al., 2017) and are based on either asset-level data or data for each component of an asset that is aggregated to obtain asset-level curves (Gentile et al., 2022). Flood vulnerability curves are often based on expert judgement supported with (little) empirical data (Kok et al., 2005; Vanneuville et al., 2006; Vrisou van Eck and Kok, 2001) and are thus more generalized in nature (Gentile et al., 2022). We find flood curves for three levels of detail. First, the (highly) 'generalized' curves are assumed to be a representative for multiple but highly diverse infrastructure types. For example, the 'infrastructure' curves provided by (Huizinga et al., 2017) that are developed for (coarse) grid-based modelling of the flood risk. Such curves are useful for gaining an impression of the total infrastructure damage of large-scale flood events (e.g. national or continental scale risk assessments), but one cannot expect them to give accurate results for single assets and in detailed studies, as demonstrated by Jongman et al., (2012) and Van Ginkel et al. (2021). Second, 'joint' curves are often assumed to be applicable to multiple types of infrastructures that have similar physical characteristics. For example, the Kok et al. (2005) curve for electricity systems is also used for the communication system, and the curve for roads is also used for railways. Third, 'object-based' curves represent the vulnerability of a specific infrastructure type in more detail and specifically account for structure-specific attributes (e.g., Van Ginkel et al., 2021; Kellermann et al., 2015). However, also in these studies, the curves cannot be seen in isolation from the type and resolution of the hazard model for which they were initially developed. For example, both Van Ginkel et al. (2021) and Kellermann et al. (2015) anticipate a coarse 100*100 m inundation model that cannot 'see' the local elevation of highways and rail embankments. Therefore, their vulnerability curves start from ground level, and not from the local embankment level. A high-resolution (e.g. 1*1 m) inundation model would detect this embankment level as the ground level, resulting in much lower water depths. The original vulnerability curves would therefore need to be corrected before they are used in a higher-resolution model.

## 5 Conclusion and recommendations

Through our systematic literature review, we have collected over 1,510 fragility and vulnerability curves, making it the most complete publicly available vulnerability database for CI to date. All curves have been standardized to allow for an easy starting point for any (multiple) hazard and (multi-)infrastructure risk assessment. Yet, the literature review has also highlighted that there are substantial differences in availability of curves across hazards and CI systems. Earthquakes has

received most attention and development of curves across CI systems, whereas wind curves predominantly focused on energy alone. Generally, most development has focused on energy and transportation, whereas work is still to be done on telecommunication in particular.

We have taken the opportunity to leverage upon the wealth of existing literature to develop the physical vulnerability database. Even though we have compiled this database from an extensive review, we cannot rule out that we have excluded some studies. Also, we decided to not include all of the curves that have been extensively reviewed in earlier publications. Instead, we decided to refer to the specific review and pointed out some of the key literature, such as for pipeline damage due to earthquakes. Additionally, we wish to highlight that we have not conducted a quality check of the curves, but rather focused on establishing an overview of the current literature on the curves and the collection of these for the database. When considering their usage, it is essential to also account for the resolution, adaptability, and transferability of the curves in assessing and managing risks to CI across various settings and scenarios. In supporting this, we consistently summarized characteristics of each curve in Table D1 of our database.

We strongly encourage users to expand the database with: (1) existing curves that are currently not included, (2) curves for other hazard types, such as wildfires and extreme cold, (3) curves for other important infrastructures types, such as bridges, (4) curves for various infrastructure characteristics, such as form (e.g., low-rise) and construction materials, (5) curves that consider the joint effect of multiple intensity measures of a single hazard, and (6) curves that consider the interaction of multi-hazards. Another point for future research could be developing a graphical user interface for the database, such as the one developed for GEM. Additionally, impacts from natural hazards go beyond physical damages, encompassing consequences such as repair time, operational disruptions of infrastructure and systemic vulnerability due to interdependencies. Inclusion of curves that address these consequences, including fragility curves considering the failure of an asset to continue its core function, would further enrich the database.

There are several opportunities for improved hazard vulnerability and risk assessments provided through our database. Standardising curves for hazard-asset combinations makes it easier to compare vulnerability of assets to same hazards levels around the world and investigate the underlying factors such as general construction types and asset dimensions. This will further make it easier to do a global assessment of comparative (multiple) hazard direct damage assessments across multiple infrastructure types. Our database also captures the uncertainty in several curve estimates and where such uncertainty is not provided within a certain class of hazard-asset curves, it is possible to do a sensitivity analysis of damage assessments across multiple curves. In creating this database, we have also provided a template for adding new fragility and vulnerability curves, which would help the research community to enrich this further for collaborative use.

Appendix A: Search term syntax and number of records

We used a total number of 125 search terms for this systematic literature review of which an overview of the syntax is provided in Table A1. We also provide the approximate number of records Google Scholar found for each search term syntax.

**Table A1: Overview of the search term syntax and number of records using Google scholar. Please note that Google Scholar only shows an approximate of records found.**

| Hazard type | Search term syntax | Number of records |
|---|---|---|
| Flooding | 'power plant' AND 'flooding' AND 'vulnerability curve' | 39,400 |
| | 'power' AND 'flooding' AND vulnerability curve' | 62,400 |
| | 'power' AND 'flooding' AND 'depth-damage curve' | 1,930 |
| | 'substation' AND 'flooding' AND 'vulnerability curve' | 4,880 |
| | 'substation' AND 'flood' AND 'vulnerability curve' | 4,880 |
| | 'power pole' AND 'flood' AND 'vulnerability curve' | 24,000 |
| | 'energy' AND 'flood' AND 'vulnerability curve' | 80,000 |
| | 'electricity' AND 'flood' AND 'vulnerability curve' | 31,600 |
| | 'power pole' AND 'flood' AND 'vulnerability curve' | 24,000 |
| | 'cable' AND 'flood' AND 'vulnerability curve' | 19,600 |
| | 'railway' AND 'flood' AND 'vulnerability curve' | 20,900 |
| | 'airports' AND 'flood' AND 'vulnerability curve' | 18,600 |
| | 'airports' AND 'flood' AND 'depth-damage curve' | 329 |
| | 'telecommunication' AND 'flood' AND 'depth-damage curve' | 1,330 |
| | 'telecommunication' AND 'flood' AND 'vulnerability curve' | 18,000 |
| | 'telecommunication' AND 'flood' AND 'fragility curve' | 8,140 |
| | 'hospital' AND 'flood' AND 'vulnerability curve' | 25,900 |
| | 'hospital' AND 'flood' AND 'depth-damage function' | 567 |
| | 'hospital' AND 'flooding' AND 'depth-damage function' | 567 |
| | 'health' AND 'flood' AND 'depth-damage function' | 1,750 |
| | 'education' AND 'flood' AND 'depth-damage function' | 2,260 |
| | 'school' AND 'flood' AND 'depth-damage function' | 1,880 |
| | 'water' AND 'flood' AND 'depth-damage function' | 3,490 |
| | 'water' AND 'flood' AND 'vulnerability curve' | 130,000 |
| | 'water well' AND 'flood' AND 'vulnerability curve' | 142,000 |
| | 'water treatment plant' AND 'flood' AND 'vulnerability curve' | 36,100 |
| | 'storage tank' AND 'flood' AND 'vulnerability curve' | 23,100 |
| | 'transmission pipeline' AND 'flood' AND 'vulnerability curve' | 18,600 |
| | 'water tower' AND 'flood' AND 'vulnerability curve' | 23,200 |
| | 'water works' AND 'flood' AND 'vulnerability curve' | 68,400 |
| | 'wastewater treatment plant' AND 'flood' AND 'vulnerability curve' | 20,400 |
| | 'waste transfer station' AND 'flood' AND 'vulnerability curve' | 22,700 |
| | 'waste' AND 'flood' AND 'vulnerability curve' | 37,300 |
| | 'roads' AND 'flooding' AND 'vulnerability curve' | 34,800 |
| | 'roads' AND 'flooding' AND 'fragility curve' | 19,300 |
| Earthquakes | 'power' AND 'earthquake' AND vulnerability curve' | 43,800 |
| | 'power' AND 'earthquake' AND 'fragility curve' | 26,100 |
| | 'substation' AND 'earthquake' AND 'fragility curve' | 2,310 |

| | | |
|---|---|---:|
| | 'pole' AND 'earthquake' AND 'fragility curve' | 11,700 |
| | 'energy' AND 'earthquake' AND 'fragility curve' | 30,300 |
| | 'cable' AND 'earthquake' AND 'fragility curve' | 12,000 |
| | 'tower' AND 'earthquake' AND 'fragility curve' | 16,000 |
| | 'telecommunication' AND 'earthquake' AND 'fragility curve' | 4,730 |
| | 'telecommunication' AND 'earthquake' AND 'vulnerability curve' | 11,000 |
| | 'mast' AND 'earthquake' AND 'vulnerability curve' | 4,030 |
| | 'communication tower' AND 'earthquake' AND 'vulnerability curve' | 19,900 |
| | 'water' AND 'earthquake' AND 'fragility curve' | 25,500 |
| | 'water well' AND 'earthquake' AND 'fragility curve' | 25,700 |
| | 'water treatment plant' AND 'earthquake' AND 'fragility curve' | 17,500 |
| | 'storage tank' AND 'earthquake' AND 'fragility curve' | 17,400 |
| | 'water transmission pipe' AND 'earthquake' AND 'fragility curve' | 16,800 |
| | 'water tower' AND 'earthquake' AND 'fragility curve' | 17,900 |
| | 'water works' AND 'earthquake' AND 'fragility curve' | 21,700 |
| | 'waste water treatment plant' AND 'earthquake' AND 'fragility curve' | 17,100 |
| | 'waste transfer station' AND 'earthquake' AND 'fragility curve' | 16,600 |
| | 'waste' AND 'earthquake' AND 'fragility curve' | 16,200 |
| | 'hospital' AND 'earthquake' AND 'vulnerability curve' | 20,100 |
| | 'hospital' AND 'earthquake' AND 'fragility curve' | 15,900 |
| | 'health facility' AND 'earthquake' AND 'fragility curve' | 18,300 |
| | 'education' AND 'earthquake' AND 'fragility curve' | 20,100 |
| | 'school' AND 'earthquake' AND 'fragility curve' | 22,700 |
| | 'airports' AND 'earthquake' AND 'fragility curve' | 8,320 |
| | 'runways' AND 'earthquake' AND 'fragility curve' | 6,430 |
| | 'railway' AND 'earthquake' AND 'vulnerability curve' | 18,700 |
| Windstorms | 'power' AND 'wind' AND vulnerability curve' | 106,000 |
| | 'substation' AND 'wind' AND 'fragility curve' | 2,990 |
| | 'pole' AND 'wind' AND 'fragility curve' | 19,300 |
| | 'cable' AND 'wind' AND 'fragility curve' | 18,400 |
| | 'tower' AND 'wind' AND 'fragility curve' | 20,300 |
| | 'energy' AND 'hurricane' AND 'fragility curve' | 17,800 |
| | 'roads' AND 'hurricane' AND 'vulnerability curve' | 21,200 |
| | 'runways' AND 'wind' AND 'fragility curve' | 6,780 |
| | 'airports' AND 'wind' AND 'fragility curve' | 16,900 |
| | 'railway' AND 'wind' AND 'fragility curve' | 17,800 |
| | 'telecommunication' AND 'hurricane' AND 'vulnerability curve' | 7,260 |
| | 'telecommunication' AND 'hurricane' AND 'fragility curve' | 2,880 |
| | 'water' AND 'cyclones' AND 'fragility curve' | 12,600 |
| | 'water infrastructure' AND 'cyclones' AND 'fragility curve' | 16,800 |
| | 'water well' AND 'cyclones' AND 'fragility curve' | 17,900 |
| | 'water well' AND 'cyclones' AND 'vulnerability curve' | 27,900 |
| | 'water treatment plant' AND 'cyclones' AND 'fragility curve' | 16,500 |
| | 'storage tank' AND 'wind' AND 'fragility curve' | 18,200 |
| | 'water transmission pipe' AND 'hurricane' AND 'fragility curve' | 16,400 |
| | 'water tower' AND 'hurricane' AND 'fragility curve' | 15,500 |
| | 'water works' AND 'hurricane' AND 'fragility curve' | 18,200 |
| | 'wastewater treatment plant' AND 'hurricane' AND 'fragility curve' | 6,400 |

| | | |
|---|---|---:|
| | 'waste' AND 'hurricane' AND 'fragility curve' | 13,000 |
| | 'hospital' AND 'hurricane' AND 'fragility curve' | 11,300 |
| | 'hospital' AND 'wind' AND 'fragility curve' | 20,600 |
| | 'health facility' AND 'wind' AND 'fragility curve' | 20,800 |
| | 'education' AND 'wind' AND 'fragility curve' | 26,300 |
| Landslides | 'power' AND 'landslide' AND vulnerability curve' | 21,400 |
| | 'substation' AND 'landslide' AND 'vulnerability curve' | 1,120 |
| | 'pole' AND 'landslide' AND 'vulnerability curve' | 7,070 |
| | 'cable' AND 'landslide' AND 'vulnerability curve' | 6,170 |
| | 'tower' AND 'landslide' AND 'fragility curve' | 4,110 |
| | 'energy' AND 'landslide' AND 'vulnerability curve' | 21,900 |
| | 'roads' AND 'landslide' AND 'vulnerability curve' | 19,900 |
| | 'runways' AND 'landslide' AND 'vulnerability curve' | 1,220 |
| | 'runways' AND 'landslide' AND 'fragility curve' | 2,060 |
| | 'airports' AND 'landslide' AND 'fragility curve' | 2,920 |
| | 'railway' AND 'landslide' AND 'fragility curve' | 5,210 |
| | 'telecommunication' AND 'landslide' AND 'vulnerability curve' | 4,080 |
| | 'telecommunication' AND 'landslide' AND 'fragility curve' | 1,700 |
| | 'water' AND 'landslide' AND 'vulnerability curve' | 26,200 |
| | 'water infrastructure' AND 'landslide' AND 'vulnerability curve' | 21,000 |
| | 'water well' AND 'landslide' AND 'vulnerability curve' | 26,400 |
| | 'water treatment plant' AND 'landslide' AND 'vulnerability curve' | 18,700 |
| | 'storage tank' AND 'landslide' AND 'vulnerability curve' | 16,400 |
| | 'water transmission pipe' AND 'landslide' AND 'vulnerability curve' | 17,500 |
| | 'water tower' AND 'landslide' AND 'vulnerability curve' | 17,100 |
| | 'water works' AND 'landslide' AND 'vulnerability curve' | 22,200 |
| | 'wastewater treatment plant' AND 'landslide' AND 'vulnerability curve' | 12,400 |
| | 'waste' AND 'landslide' AND 'vulnerability curve' | 14,500 |
| | 'hospital' AND 'landslide' AND 'fragility curve' | 4,710 |
| | 'health facility' AND 'landslide' AND 'fragility curve' | 14,100 |
| | 'education' AND 'wind' AND 'fragility curve' | 26,300 |
| General | 'natural disaster' AND 'critical infrastructure' AND 'vulnerability curve' | 61,700 |
| | 'natural disaster' AND 'critical infrastructure' AND 'fragility curve' | 24,800 |
| | 'natural hazard' AND 'critical infrastructure' AND 'vulnerability curve' | 59,100 |
| | 'natural hazard' AND 'critical infrastructure' AND 'fragility curve' | 26,100 |
| | 'natural disaster' AND 'lifeline' AND 'vulnerability curve' | 24,300 |
| | 'natural disaster' AND 'lifeline' AND 'fragility curve' | 19,900 |
| | 'natural hazard' AND 'lifeline' AND 'vulnerability curve' | 23,300 |
| | 'natural hazard' AND 'lifeline' AND 'fragility curve' | 21,300 |

*Data availability.* The dataset for multiple hazard fragility and vulnerability curves is publicly available through the following Zenodo repository: https://zenodo.org/doi/10.5281/zenodo.10203845.


*Author contributions.* S.N. conceived and designed the research with contributions of E.E.K., P.J.W., and J.C.J.H.A. S.N. developed the methodology, conducted the literature review, and collected the vulnerability and fragility curves into one

harmonized database. K.C.H.G. and M.Y. contributed to the literature review and M.Y. also contributed to the collecting of vulnerability and fragility curves. All authors contributed to the writing of the manuscript.


*Competing interests.* One of the co-authors, Philip Ward, is member of the editorial board of Natural Hazards and Earth System Sciences.

*Financial support.* This research is part of the project Remote Climate Effects and their Impact on European sustainability,
Policy and Trade (RECEIPT) that is supported by the European Union's Horizon 2020 Research and Innovation Programme under grant agreement no. 820712. E.E.K. and P.J.W. were further supported by the Dutch Research Council (NWO) (grant nos. VI.Veni.194.033 and VI.Vidi.016.161.324), the EU-H2020 CoCliCo project (grant no. 101003598) and MYRIAD-EU (European Union's Horizon 2020 research and innovation programme under grant no. 101003276). M.Y. was funded by the China Scholarship Council (CSC). J.C.J.H.A. was supported by the VICI grant no. 453-13-006, and ERC advanced grant
884442. E.E.K., K.C.H.G. and R.P. received funding from EU-H2020 MIRACA, grant no. 101093854. R.P. also received funding from the Climate Compatible Growth (CCG) project of the UK Foreign, Commonwealth and Development Office.

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
