# Peer review of "Review article: Physical Vulnerability Database for Critical Infrastructure Hazard Risk Assessments – A systematic review and data collection"

_Natural Hazards and Earth System Sciences, 2023_

## Referee Comment (RC1)

**Review article: Physical Vulnerability Database for Critical Infrastructure Multi-Hazard Risk Assessments – A systematic review and data collection**

This manuscript provides a systematic literature review on - and accompanying extensive database of - fragility and vulnerability curves for different types of critical infrastructure and hazards. I commend the authors for their significant efforts in creating this review and developing the associated database that will undoubtedly prove helpful for various risk modelling applications. Furthermore, I believe that the manuscript would be of interest to readers of this journal. However, I provide below a number of comments that I think should be addressed before the manuscript could be published.

**Main comments:**
1. While I appreciate the authors' efforts in developing such a comprehensive database and review, there are certain (sometimes questionable) limitations that need to be more explicit in the text, as well as some ambiguity on its scope that needs to be removed:
    a. The review does not mention multi-hazard vulnerability curves. These curves either account for one intensity measure per hazard or assume a state-dependent format that tracks the evolving vulnerability of the asset as hazard events occur. (Please see Section 2.1.4 of https://doi.org/10.1016/j.ijdrr.2022.103365 for a more thorough description of these types of curves). Given the ever-increasing focus in the field of risk analysis on multi-hazards, this seems like quite a substantial oversight and I would like the authors to at least justify this decision.
    b. It is not clear from the review that single-hazard vulnerability curves may also use more than one intensity measure. (For instance, see: https://doi.org/10.1016/j.ress.2020.106971, which presents flood fragility functions that make use of both flood depth and flood duration information). It is not clear whether such vector-valued approaches were considered in this review.
    c. The review does not explicitly cover damage-to-loss (or, more broadly, "damage-to-impact") models. (Please see Section 2.1.3 of https://doi.org/10.1016/j.ijdrr.2022.103365 or https://doi.org/10.1002/eqe.2687, for instance). These types of models act as a crucial link between fragility and vulnerability curves (as implied by equation 1), and therefore it should be made clear that they are outside the scope of this specific review. (I understand from line 181 that information on these models is made available in the database if they are included within the original source of a fragility/vulnerability curve, but they are not reviewed explicitly as a separate entity).
    d. It is important to note that while vulnerability curves can be used to capture losses for individual critical infrastructure *components* (nodes or assets), they are not the only ingredient necessary to quantify losses and damage for distributed critical infrastructure *systems*. The vulnerability analysis also needs to account for interdependencies between the multiple components that underly the system as well as the individual functionalities of each component, for instance.
    e. When it comes to curves for buildings in particular (under the health and education category), it is not clear what the review covers in terms of buildinglevel versus component-level resolution and structural versus non-structural damages.

 f. It is surprising that there is no section devoted to the definition of each type of critical infrastructure considered in the review (i.e., what types of underlying components are assumed to comprise each), to clarify the scope of the various subcomponents intended to be captured by the database. Currently, it is not clear whether the absence of some infrastructure subcomponents from Section 3 is due to the lack of availability of curves or whether these types of subcomponents were deemed to be outside the scope of the review.

 g. I may have missed it, but I could not find any details on the years of publication covered by the review.

2. There is various terminology used throughout the manuscript that requires more accurate, rigorous treatment:

 a. Line 25: the phrase "at risk to natural hazards" does not make sense, since a hazard is itself a fundamental underlying component of risk

 b. Line 40: To avoid confusion around the differences between fragility and vulnerability functions, the output of a vulnerability function should be referred to as "loss" rather than "damage". This would avoid confusion between the terms "damage factor" (that is used to describe the output of a vulnerability curve at line 39) and "damage state" (that is used when describing the output of a fragility function at line 41). It is important to clearly distinguish between the terms "damage" and "loss"; they are not equivalent. Fragility functions provide a prediction of a binary outcome related to some binary failure criterion (e.g., collapse/no collapse, breach/no breach, breakage/no breakage), whereas vulnerability curves quantify some continuous consequence outcome (e.g., repair cost, repair time).

 c. It is important to specify that the term "cost" as used in this context is not limited to monetary cost; vulnerability curves can be used to measure repair time, for instance . It would be more appropriate to use a term like "consequence".

 d. I think it is important that vulnerability "data" is not conflated with vulnerability "curves" – data are used to fit the curves, thus the two are not equivalent. Furthermore, I would suggest describing them as "curves" or "functions" (e.g., as at line 57) but not both.

 e. Line 122: It is not clear to me what is meant by the term "risk indicator".

3. For a systematic review, I find Section 3 to be surprisingly disorganised. There does not appear to be a consistent structure used for any of the subsections. For instance:

 a. A definition of transport is provided in Section 3.1.2, but no such definition is provided for energy in Section 3.1.1.

 b. The terms "damage function" (line 236) and "depth-damage function" (line 244) are both used in Section 3.1.2, but it is not clear what the distinction between these two functions is.

 c. There does not seem to be any systematic division between discussions on vulnerability curves and fragility curves in any given subsection

 d. The types of components covered for each critical infrastructure are not consistent across the different hazards. For instance, telecommunications covers broad "communication systems" for flooding, whereas it covers

"monopile towers", "central offices" and "broadcasting stations" for earthquakes.

    e. Studies and their associated curves are described to various degrees of detail, even within a given subsection. For instance in Section 3.1.1, we are told that a fragility curves for certain transmission towers were developed "using a reliability analysis based on kriging", yet in the same section we are told that another study "evaluated the fragility of a transmission tower-line system" with no more details provided.

The lack of organisation and consistency across Section 3 makes the section difficult to read and digest. I would suggest that the section could be improved through the use of tables that summarise pertinent information on the curves for different critical infrastructure and hazards (e.g., subcomponent that it relates to, country of origin, method of development, types of losses/damage captured, hazard intensity measure, other important specific notes) in a consistent manner. Furthermore, it would be useful if each subsection was consistently divided into a separate discussions on fragility curves and vulnerability curves.

**More minor comments:**

1. Given the nature of the paper, I think the abstract could benefit from an explicit definition of the term "vulnerability"
2. Equation 1: It seems a bit strange to use "E" to denote a probability when this variable is normally reserved for referring to expected values. I would suggest that the authors either formulate equation 1 as an expected (or mean) value (using "E") or a probability of exceedance using "p" instead of "E".
3. I believe that the term "intensity measure" is more common in the literature than "intensity metric" and the authors might want to consider switching to the former.
4. Line 43-45: I think the statement made here about fragility versus vulnerability curves incorrectly implies that vulnerability curves are not used extensively in earthquake risk analyses (e.g., see https://doi.org/10.1193/011816EQS015DP)
5. Line 55: "limited represented"- fix the grammar here
6. Line 160: Are the cost values country specific? If so, this should be made clear in the database.
7. Section 2.2: Please clarify if the database contains information on the mathematical form of each curve and the associated parameters.
8. The conclusions section mentions that the authors encourage more contributions to the database "for a wide range of building types". But this would not necessarily comply with the critical infrastructure focus of the database.
9. The DOI provided for the database in the preprint did not work when I clicked on it. I used the DOI provided in the manuscript review centre instead, so I am not sure if there is any error with the address provided in the preprint.

---

## Author Comment (AC1)

**Review article: Physical Vulnerability Database for Critical Infrastructure Multi-Hazard Risk Assessments – A systematic review and data collection**
**Authors: Sadhana Nirandjan, Elco E. Koks, Mengqi Ye, Raghav Pant, Kees C.H. Van Ginkel, Jeroen C.J.H. Aerts, and Philip J. Ward**

*Response to Anonymous Referee #1*

This manuscript provides a systematic literature review on - and accompanying extensive database of - fragility and vulnerability curves for different types of critical infrastructure and hazards. I commend the authors for their significant efforts in creating this review and developing the associated database that will undoubtedly prove helpful for various risk modelling applications. Furthermore, I believe that the manuscript would be of interest to readers of this journal. However, I provide below a number of comments that I think should be addressed before the manuscript could be published.

*[Authors' reply] We thank the reviewer for the positive remarks and for recognizing our efforts in creating this manuscript and developing the database. We are pleased that the reviewer recommends publication in Natural Hazards and Earth System Sciences following a revision of major and minor comments. We have addressed these points in the revised manuscript, and agree that this has led to a further improvement of the manuscript.*

**Main comments**
1. While I appreciate the authors' efforts in developing such a comprehensive database and review, there are certain (sometimes questionable) limitations that need to be more explicit in the text, as well as some ambiguity on its scope that needs to be removed:
   a) The review does not mention multi-hazard vulnerability curves. These curves either account for one intensity measure per hazard or assume a state-dependent format that tracks the evolving vulnerability of the asset as hazard events occur. (Please see Section 2.1.4 of https://doi.org/10.1016/j.ijdrr.2022.103365 for a more thorough description of these types of curves). Given the ever-increasing focus in the field of risk analysis on multi-hazards, this seems like quite a substantial oversight and I would like the authors to at least justify this decision.

   *[Authors' reply] Thank you for your comment. In fact, we touch upon multi-hazard vulnerability curves sourced from three references in Section 3 and shortly mention these multi-hazard curves in our discussion. We did not find any other multi-hazard curves related to critical infrastructure. We were not familiar with the terminology as used in the paper highlighted by reviewer #1. Using this knowledge about the mathematical definitions, we have made multiple revisions for consistency and clarification purposes.*

   *We have decided to add the state-dependent curves to our database as they presume a form of precondition. As explained in Section 2.2, we identify 'additional characteristics' that should specifically be mentioned if these characteristics are fundamental for the vulnerability of a given infrastructure asset. We have added the following line to the description of 'Additional characteristics' (lines 137-138):*

   > *'Please note that we also provide details if a curve incorporated conditions that are sustained from a previous hazard.'*

*Furthermore, we already included the state-dependent curves as presented by Teoh et al. (2019), but only included the curves without a precondition (or primary hazard) in our review of Argyroudis et al. (2018). An additional 24 fragility curves, which are state-dependent curves, have now been incorporated into the database, and we have adjusted the following lines in Section 3 (lines 430-433):*

> *'Argyroudis et al. (2018) apply an analytical approach to develop fragility curves for three damage states ('minor', 'moderate', and 'extensive/complete') for the representation of highways [E7.1-5] and railways [E8.11-15] on an embankment. They analysed the joint effect of flooding and earthquake by using a range of inundation depths as a precondition for their model.'*

*Moreover, in the case of Teoh et al. (2019), we have now also explicitly added the precondition (i.e., icing conditions leading to additional load on conductors and neutral wire) to the database's Table D1 'Summary CI vulnerability data' under 'Additional characteristics'.*

*Lastly, we decided to not include vector-valued dual hazard curves that are described by multi-variate distribution functions in our database due to the complexity of replicating them. Moreover, we only found one such curve in the current literature. Lines 645-649 now read:*

> *'Zhu et al. (2022) assessed the vulnerability of roads to tropical cyclones and their joint effect of precipitation and wind speed by using damage records from events in Hainan Province, China. These records include damage observations to various structures such as protection components of a road, pavement and subgrade. In our database, we include the physical damage probability curve that applies the maximum wind speed at 10 m above ground-level as the intensity measure [W7.1] and refer to Zhu et al. (2022) for the multi-variate curve for the concurrent compound hazard intensities.'*

*However, we touch upon the multi-hazard curves as a recommendation for future study in Section 4 (see our recommendations 5 and 6 under reviewer's comment 1b).*

b) It is not clear from the review that single-hazard vulnerability curves may also use more than one intensity measure. (For instance, see: https://doi.org/10.1016/j.ress.2020.106971, which presents flood fragility functions that make use of both flood depth and flood duration information). It is not clear whether such vector-valued approaches were considered in this review.

*[Authors' reply] Thanks for the interesting literature demonstrating multi-variate models for buildings. We do not include multi-variate models and would like to refer to our response on comment 1a of reviewer #1. Additionally, we have added this topic as a subject for future study (recommendation 5 in lines 876-881):*

> *'We strongly encourage users to expand the database with: (1) existing curves that are currently not included, (2) curves for other hazard types, such as wildfires and extreme cold, (3) curves for other important infrastructures types, such as bridges, (4) curves for various building typologies with regard to form (e.g., low-rise) and construction materials,*

> *(5) curves that consider the joint effect of multiple intensity measures of a single hazard, and (6) curves that consider the interaction of multi-hazards.'*

c) The review does not explicitly cover damage-to-loss (or, more broadly, "damage-to-impact") models. (Please see Section 2.1.3 of https://doi.org/10.1016/j.ijdrr.2022.103365 or https://doi.org/10.1002/eqe.2687, for instance). These types of models act as a crucial link between fragility and vulnerability curves (as implied by equation 1), and therefore it should be made clear that they are outside the scope of this specific review. (I understand from line 181 that information on these models is made available in the database if they are included within the original source of a fragility/vulnerability curve, but they are not reviewed explicitly as a separate entity).

*[Authors' reply] Thank you for raising damage-to-loss models and associated literature to our awareness. We indeed do not review the damage-to-loss models as a separate entity in our study, but use them to convert fragility curves to vulnerability curves if provided in the original source of the curves. However, we do not provide the parameters of the damage-to-loss models in our database, but refer to the original source of the curves instead. To make more clear how we included damage-to-loss models, we have added the following to section 2.3 (lines 195-198):*

> *'Damage-to-loss models, also known as damage-to-impact or consequence models, act as a crucial link between fragility and vulnerability curves by relating physical damage with a damage or loss metric (Martins et al., 2016; Yepes-Estrada et al., 2016; Gentile et al., 2022). A review of these damage-to-loss models, however, is outside the scope of this study.'*

d) It is important to note that while vulnerability curves can be used to capture losses for individual critical infrastructure *components* (nodes or assets), they are not the only ingredient necessary to quantify losses and damage for distributed critical infrastructure *systems*. The vulnerability analysis also needs to account for interdependencies between the multiple components that underly the system as well as the individual functionalities of each component, for instance.

*[Authors' reply] We acknowledge that vulnerability curves are not the only ingredient necessary for the quantification of damages and losses for distributed infrastructure systems. Systemic vulnerability, originating from the mutual linkages/interactions among physical, economic, and social systems, is outside the scope of this paper. In our introduction, we put the definition of vulnerability and fragility curves in context of physical asset damages and we explain that the result of our study can be used to assess direct physical asset damages to CI. We have now also edited lines 48-50 to emphasize that we focus on physical asset damages:*

> *'While researchers have made significant progress in the development of fragility and vulnerability curves focusing on physical damages of different CI assets due to various natural hazards, no study has yet combined these curves into one extensive (multiple hazard) database for CI.'*

*Furthermore, in the revised version of our paper, we now make reference to interdependencies in Section 5 with the following text (lines 881-884):*

> *'Additionally, impacts from natural hazards go beyond physical damages, encompassing consequences such as repair time, operational disruptions of infrastructure and systemic vulnerability due to interdependencies. Inclusion of curves that address these consequences, including fragility curves considering the failure of an asset to continue its core function, would further enrich the database.'*

e) When it comes to curves for buildings in particular (under the health and education category), it is not clear what the review covers in terms of building-level versus component-level resolution and structural versus non-structural damages.

   *[Authors' reply] Thank you for your comment. The curves that we found for health and education buildings are provided at building-level. Any important characteristics of health and education buildings are summarized in the database's table D1 'Summary CI Vulnerability Data' under the column 'Additional characteristics'.*

f) It is surprising that there is no section devoted to the definition of each type of critical infrastructure considered in the review (i.e., what types of underlying components are assumed to comprise each), to clarify the scope of the various subcomponents intended to be captured by the database. Currently, it is not clear whether the absence of some infrastructure subcomponents from Section 3 is due to the lack of availability of curves or whether these types of subcomponents were deemed to be outside the scope of the review.

   *[Authors' reply] Thanks. We follow the CI categorization presented by Nirandjan et al. (2022) as explained in the methodology and provide an overview of the infrastructure types included in Table 1. Additionally, we have now added figure 2 to Section 3 which provides an overview of different variables per infrastructure type and hazard type combination. At the same time, this figure also clarifies which infrastructure types were included in the study.*

g) I may have missed it, but I could not find any details on the years of publication covered by the review.

   *[Authors' reply] Thank you for pointing this out. We conducted our systematic review without confining ourselves to a specific search window and have now made this explicit in lines 105-106:*

   > *'However, we did not limit the search window and the geographical scope of the study and are thus still able to provide insight into vulnerability curves in various contexts.'*

2. There is various terminology used throughout the manuscript that requires more accurate, rigorous treatment:
   a) Line 25: the phrase "at risk to natural hazards" does not make sense, since a hazard is itself a fundamental underlying component of risk

   *[Authors' reply]. Hazard is indeed a fundamental underlying component of the risk equation that also includes exposure and vulnerability. However, the phrase "at risk to" is used to convey the*

*idea that individuals, assets or entities are exposed to the possibility of harm or damage resulting from natural hazards. It implies that there is a potential for adverse consequences to occur as a result of the presence or occurrence of natural hazards in their vicinity.*

b) Line 40: To avoid confusion around the differences between fragility and vulnerability functions, the output of a vulnerability function should be referred to as "loss" rather than "damage". This would avoid confusion between the terms "damage factor" (that is used to describe the output of a vulnerability curve at line 39) and "damage state" (that is used when describing the output of a fragility function at line 41). It is important to clearly distinguish between the terms "damage" and "loss"; they are not equivalent. Fragility functions provide a prediction of a binary outcome related to some binary failure criterion (e.g., collapse/no collapse, breach/no breach, breakage/no breakage), whereas vulnerability curves quantify some continuous consequence outcome (e.g., repair cost, repair time).

*[Authors' reply] We agree that a clear distinction needs to be made between 'damage,' which refers to the physical harm or deterioration suffered by infrastructure or assets, and 'loss,' which encompasses the broader consequences, including economic, social, and environmental impacts resulting from that damage. Throughout our abstract, introduction and methodology, we clearly explain that we only consider curves that specifically consider physical damages. We do not include vulnerability curves considering repair time or another form of loss, but rather focused on physical damages of an asset (expressed in damage factors in relation to an intensity measure). We therefore do not refer to 'loss' for the output of a vulnerability function but to damage.*

c) It is important to specify that the term "cost" as used in this context is not limited to monetary cost; vulnerability curves can be used to measure repair time, for instance. It would be more appropriate to use a term like "consequence".

*[Authors' reply] We agree that the impacts are not limited to monetary costs. In this study, we are interested in curves that specifically consider damages to assets which can be used in damage risk assessments to assess the physical damage caused by flooding, earthquakes, windstorms and landslides. We therefore do not use the term consequence as this would imply that we would also include metrics such as repair time. We have made a slight adjustment in the abstract to better convey this (lines 18-20):*

> *'The publicly available centralized database that contains over 1,510 curves can directly be used as input for risk assessment studies that evaluate the potential physical damages to assets due to flooding, earthquakes, windstorms and landslides.'*

*Furthermore, in the original manuscript we consistently use the term 'physical damage' to indicate that we focus on the damage to an infrastructure asset caused by a natural hazard. Also, following reviewer's #1 and #2 comments on the methodology, we have improved the methodology's section which now also explicitly mentions the inclusion and exclusion criteria, from which becomes clear that we only focus on physical damages on infrastructure.*

*However, we now have edited the last sentence of third paragraph in the 'Conclusion and recommendations' section to highlight that impacts from natural hazards go beyond asset damages (lines 881-884):*

> *'Additionally, impacts from natural hazards go beyond physical damages, encompassing consequences such as repair time and operational disruptions of infrastructure. Inclusion of curves that address these consequences, including fragility curves considering the failure of an asset to continue its core function, would further enrich the database.'*

d) I think it is important that vulnerability "data" is not conflated with vulnerability "curves" – data are used to fit the curves, thus the two are not equivalent. Furthermore, I would suggest describing them as "curves" or "functions" (e.g., as at line 57) but not both.

*[Authors' reply] Thank you for pointing this out. We have now eliminated the usage of vulnerability 'data' and use 'curves' instead. However, we find 'vulnerability data' an appropriate term to use in section 4 in which we compare the collected data within our database across hazard types and CI types. This data encompasses the curves as well as the wide range of characteristics that we have collected for our database as outlined in section 2.2.*

*Furthermore, we have replaced 'functions' with 'curves' for consistency purposes. For flood section 3.1 in particular, we often used 'depth-damage function' and 'damage function' in the original manuscript and have now replaced these with 'depth-damage curve'. We also provide a definition of depth-damage curves in lines 231-232:*

> *'FEMA (2013) developed depth-damage curves (i.e., vulnerability curves that relate the flood inundation depth to the potential physical damage), for power plants with varying capacities for the US …'*

e) Line 122: It is not clear to me what is meant by the term "risk indicator".

*[Authors' reply] We agree that the term 'risk indicator' is not clear. We have now omitted this term as we don't use this term in the remainder of the manuscript and only used it to refer to the hazard, exposure and vulnerability characteristics as discussed in section 2.2. We replaced the term 'risk indicators' with 'hazard, exposure and vulnerability' in the first paragraph of section 3.*

3. For a systematic review, I find Section 3 to be surprisingly disorganised. There does not appear to be a consistent structure used for any of the subsections. For instance:
   a) A definition of transport is provided in Section 3.1.2, but no such definition is provided for energy in Section 3.1.1.

   *[Authors' reply] We indeed did not provide a definition for energy and for other systems except transport. The underlying data that Huizinga et al. (2017) used to construct the 'transport' and 'infrastructure' curve and for what infrastructure assets these curves are applied required clarification. We therefore added definitions following Huizinga et al. (2017) to explain on what elements the 'transport' and the 'infrastructure' curves are based. Subsequently, we highlight that the 'infrastructure' curve is explicitly derived for road infrastructure and that we adapt this curve as one of the road curves in our database.*

b) The terms "damage function" (line 236) and "depth-damage function" (line 244) are both used in Section 3.1.2, but it is not clear what the distinction between these two functions is.

*[Authors' reply] Thank you for pointing this out. We used the terms 'damage function' and 'depth-damage function' interchangeably and have now made adjustments to improve consistency throughout this section. We use depth-damage curves when we specifically consider vulnerability curves that relate the flood inundation depth to the potential physical damage.*

c) There does not seem to be any systematic division between discussions on vulnerability curves and fragility curves in any given subsection

*[Authors' reply] We decided to not dedicate separate subsections to fragility curves and vulnerability curves for the following two reasons. Firstly, we did not find fragility and vulnerability curves for all asset-hazard combinations. Systematically adding separate subsections for fragility and vulnerability curves would therefore not be justified. Secondly, discussing fragility and vulnerability curves in separate sections would also cause a repetition of information, especially if curves are retrieved from the same reference. Instead, we decided to discuss the curves per reference. This may include a discussion of both fragility and vulnerability curves if both are presented in a single reference.*

*However, with the addition of the new figure 2, we also aimed to provide a reading guide which should make clear whether the reader can expect a discussion of vulnerability curves, fragility curves, both or none for specific-hazard combinations throughout sections 3.1-3.4.*

d) The types of components covered for each critical infrastructure are not consistent across the different hazards. For instance, telecommunications covers broad "communication systems" for flooding, whereas it covers "central offices" and "broadcasting stations" for earthquakes.

*[Authors' reply] Thanks. We realize that this may be interpreted as an inconsistency throughout the sections. To better guide the reader, we have added figure 2 that shows which infrastructure our review encompasses, and for which asset-hazard combinations curves were found and are thus discussed in our review.*

e) Studies and their associated curves are described to various degrees of detail, even within a given subsection. For instance in Section 3.1.1, we are told that a fragility curves for certain transmission towers were developed "using a reliability analysis based on kriging", yet in the same section we are told that another study "evaluated the fragility of a transmission tower-line system" with no more details provided.

*[Authors' reply] Thank you for pointing this out. In the revised manuscript, we have omitted the additional description 'using a reliability analysis based on kriging'. We have also screened Section 3.1-3.4 to improve the consistency in terms of the level of detail that we provide in our description for the curves. Furthermore, we now added lines 218-222 to emphasize that we were not able to*

*discuss all curves extensively in our review but rather highlighted the curves that are available and are included in our database.*

> *'In consideration of the review's length, we have chosen to not to delve into detailed discussions of all hazard, exposure and vulnerability characteristics for each curve. Instead, we focused on offering a concise overview of the current vulnerability literature in this section, whilst a complete overview of the hazard, exposure and vulnerability characteristics as discussed in section 2.2 can be found in Table D1 of our dataset.'*

The lack of organisation and consistency across Section 3 makes the section difficult to read and digest. I would suggest that the section could be improved through the use of tables that summarise pertinent information on the curves for different critical infrastructure and hazards (e.g., subcomponent that it relates to, country of origin, method of development, types of losses/damage captured, hazard intensity measure, other important specific notes) in a consistent manner. Furthermore, it would be useful if each subsection was consistently divided into a separate discussions on fragility curves and vulnerability curves.

*[Authors' reply] Thank you for your comment. We have picked up the previous comments of reviewer #1 to improve Section 3 in terms of organization and consistency. To help guide the reader throughout the Section 3, we have now added figure 2 to the beginning of Section 3 in the revised manuscript. A consistent table containing pertinent information on the curves for different critical infrastructure and hazards is provided in database's table D1 'Summary CI Vulnerability Data' containing approximately 730 rows. Since the collected data is to extensive to present for each curve in a table in the manuscript itself, we provided summary figures in Section 4 in the original manuscript. However, figures 3 and 4 lack more detailed information. Therefore, we have added figure 2 that contains information on (1) the number of curves, (2) the number of countries covered, and (3) whether there is a fragility and/or vulnerability curve available for the asset-hazard combinations considered in our review. This way, we provide a first impression of what the database covers and what is discussed in the remainder of Section 3. We have added lines 214-218 to the first paragraph of Section 3:*

> *'This section summarizes the fragility and vulnerability curves per overarching CI system, grouped in four hazard subsections (3.1 to 3.4). Figure 2 indicates the number of unique curves found in existing literature as well as the number of countries that are represented by these curves for the reviewed infrastructure-hazard combinations. Moreover, we indicate the available curve types for each infrastructure-hazard combination. The findings are provided for curves that represent infrastructure at system-level (i.e., the overarching CI systems) and asset-level (i.e., the assets that are part of the CI system).'*

*Furthermore, our choice of not discussing the fragility and vulnerability curves separately is explained under reviewer's #1 comment 3c.*

**Minor comments**
1. Given the nature of the paper, I think the abstract could benefit from an explicit definition of the term "vulnerability"

*[Authors' reply] Indeed, the abstract could benefit from an explicit definition given the nature of the paper. With regard to the count limit in the abstract, we have decided to omit the term 'vulnerability', brought the terms 'fragility curves' and 'vulnerability curves' forward and added a brief definition for both (lines 11-14):*

> *'Fragility and vulnerability curves, which quantify the likelihood of a certain damage state and the level of susceptibility of an element under varying hazard intensities, respectively, play a crucial role in comprehending, evaluating and mitigating the damages posed by natural hazards to these infrastructures.'*

2. Equation 1: It seems a bit strange to use "E" to denote a probability when this variable is normally reserved for referring to expected values. I would suggest that the authors either formulate equation 1 as an expected (or mean) value (using "E") or a probability of exceedance using "p" instead of "E".

   *[Authors' reply] We believe there may be a misinterpretation in this instance. We do not use 'E' to denote a probability. We use 'P' to denote the probability. Specifically, we use $P(ds_i|im)$ to denote the probability of an asset sustaining damage state $ds_i$ given intensity measure im. We use 'E' indeed for expected values.*

3. I believe that the term "intensity measure" is more common in the literature than "intensity metric" and the authors might want to consider switching to the former.

   *[Authors' reply] Indeed, intensity measure is more common in the literature than intensity metric. We have decided to switch to the former.*

4. Line 43-45: I think the statement made here about fragility versus vulnerability curves incorrectly implies that vulnerability curves are not used extensively in earthquake risk analyses (e.g., see https://doi.org/10.1193/011816EQS015DP)

   *[Authors' reply] Thanks. We did not want to imply that vulnerability curves are not used extensively in earthquake analyses, but rather like to highlight that the development and usage of fragility curves are well-established within the earthquake community compared to the flood community. We therefore adjusted our statement (lines 44-46):*

   > *'The development of fragility curves is a common practice within the earthquake community (Douglas, 2007), whereas the focus within the flood community tends to be on vulnerability curves (Meyer et al., 2013).'*

5. Line 55: "limited represented"- fix the grammar here

   *[Authors' reply] Whilst the grammar is not incorrect in this sentence, we have replaced 'limited' by 'inadequately' for clarity (lines 56-57):*

*'However, even within these non-public databases, CI is often inadequately represented.'*

6. Line 160: Are the cost values country specific? If so, this should be made clear in the database.

   *[Authors' reply] Yes, the cost values are country-specific. We therefore provide the country (or countries) on which the cost values in the database are based. This can be found in 'Table D3 Costs' under the column 'Geographical application'. Also, we did specify this in the original manuscript (now lines 186-191):*

   > *'Furthermore, complementary to the curve database, Table D3 contains cost numbers that can be used in combination with the curves for the estimation of potential damages, if provided in the original source from which the curve is retrieved. We indicate the infrastructure asset type, the amount and potential bandwidths, the geographical application, and on what information it is based (e.g., replacement, construction or repair costs).'*

7. Section 2.2: Please clarify if the database contains information on the mathematical form of each curve and the associated parameters.

   *[Authors' reply] The database does not contain information on the mathematical form of each curve and does not provide the associated parameters. However, we offer details about the curve's characteristics as outlined in lines 143-145. Additionally, for each curve, we specify whether parameters are presented in the original reference for the reconstruction of the curve and, indeed, references are provided for users seeking further consultation. More specifically, the information on parameters is presented in 'Table D1 Summary CI Vulnerability Data' under the column 'Readily available'. We now explicitly mention the term 'parameters' in lines 166-168 of the revised manuscript:*

   > *'Readily available. We specify whether the curve was readily available, meaning that the original source provided datapoints, parameters or a formula to reconstruct the curve. If these were not provided, we made a best estimate based on the figure to replicate the curve.'*

8. The conclusions section mentions that the authors encourage more contributions to the database "for a wide range of building types". But this would not necessarily comply with the critical infrastructure focus of the database.

   *[Authors' reply] Thank you for pointing this out. We now realize that 'building types' could also refer to a classification based on function (e.g., residential buildings). However, our intention was to focus on different building types taking into account various forms and construction materials. We have adjusted recommendation 4 accordingly. Lines 876-881 now read:*

   > *'We strongly encourage users to expand the database with: (1) existing curves that are currently not included, (2) curves for other hazard types, such as wildfires and extreme cold, (3) curves for other important infrastructures types, such as bridges, (4) curves for various building typologies with regard to form (e.g., low-rise) and construction materials, (5) curves that consider the joint effect of multiple intensity measures of a single hazard, and (6) curves that consider the interaction of multi-hazards.'*

9.  The DOI provided for the database in the preprint did not work when I clicked on it. I used the DOI provided in the manuscript review centre instead, so I am not sure if there is any error with the address provided in the preprint.

    *[Authors' reply] The DOI provided in the manuscript seem to be working for us as it directly led us to the Zenodo repository.*

---

## Author Comment (AC2)

**Review article: Physical Vulnerability Database for Critical Infrastructure Multi-Hazard Risk Assessments – A systematic review and data collection**
**Authors: Sadhana Nirandjan, Elco E. Koks, Mengqi Ye, Raghav Pant, Kees C.H. Van Ginkel, Jeroen C.J.H. Aerts, and Philip J. Ward**

*Response to Anonymous Referee #2*

This study undertakes a systematic literature review with the aim of creating a vulnerability functions database for evaluating critical infrastructure (CI) in the face of multiple hazards. While the subject matter is intriguing, the manuscript suffers from unclear methodology and lacks a clear rationale for the study. Several major concerns need addressing before considering the manuscript for publication.

*[Authors' reply] We thank the reviewer for the positive remark and are pleased that he/she recommends publication in Natural Hazards and Earth System Sciences following a further exploration of several caveats, mainly with regards our methodology and rationale for the study. We have made major and minor revisions, including a clarification of the systematic literature review, as described in the following paragraphs.*

**Major comments**

1. The study builds upon the argument of a deficiency in a centralised repository of vulnerability functions for assessing CI against multiple hazards. However, the introduction lacks a thorough examination of how this absence hinders researchers or practitioners. The discussion in the third paragraph primarily focuses on distribution and formatting issues of existing functions, neglecting considerations of resolution, adaptability, and transferability, which may have more significant impacts. It is recommended that the authors critically assess the implications of the current state of vulnerability functions

   *[Authors' reply] Thank you for your comment. Whilst we agree that considerations of resolution, adaptability and transferability are very important, the main purpose of our current manuscript is to review the status-quo of the literature on fragility and vulnerability curves for infrastructure and collect these curves for the database, rather than a quality check of the curves. In the first paragraph of our original introduction, we state the following (lines 30-32):*

   > *'Indeed, the United Nation's Sendai Framework for Disaster Risk Reduction underscores that enhanced work is needed to reduce vulnerabilities, and that freely available and accessible vulnerability information should be promoted for effective risk management (UNDRR, 2015).'*

   *With this statement, we highlight the necessity of vulnerability databases to aid effective risk management. In paragraph three, we apply this statement within critical infrastructure context to explain that a vulnerability database is lacking to date and provide examples of reasons that have led to this. However, we did adjust our statement about that the absence of a database like this hampers the research community. Lines 60-62 now read:*

   > *'A consistent overview of existing curves and an associated centralized, freely accessible database are lacking, despite the benefits they would provide to the disaster risk*

*community. These resources would enable them to perform risk assessments supported by well-informed decisions based on the current state of the fragility and vulnerability literature.'*

*With the vulnerability review and database that we provide, we support researchers and practitioners to take well-informed decisions regarding fragility and vulnerability curves. Not only do we provide the most complete publicly available vulnerability database, we have also collected characteristics for each curve, such as the derivation methodology (Supplementary Table Table_D1_Summary_CI_Vulnerability_Data). We did not perform a quality check for each curve, but we do present the curves as they are with important additional information so that this can be used in the decision-making process. We are aware of the resolution, adaptability and transferability concerns, and we provide an example of this in paragraph three of section 4.3 in our original manuscript for a specific case (lines 848-856):*

> *'Third, 'object-based' functions represent the vulnerability of a specific infrastructure type in more detail and specifically account for structure-specific attributes (e.g., Van Ginkel et al., 2021; Kellermann et al., 2015). However, also in these studies, the damage functions cannot be seen in isolation from the type and resolution of the hazard model for which they were initially developed. For example, both Van Ginkel et al. (2021) and Kellermann et al. (2015) anticipate a coarse 100\*100 m inundation model that cannot 'see' the local elevation of highways and rail embankments. Therefore, their vulnerability curves start from ground level, and not from the local embankment level. A high-resolution (e.g. 1\*1 m) inundation model would detect this embankment level as the ground level, resulting in much lower water depths. The original vulnerability curves would therefore need to be corrected before they are used in a higher-resolution model.'*

*We now also raise these concerns in our Conclusion and recommendations (Section 5) in lines 870-874 to emphasize that these need to be accounted for when applying the database:*

> *'Additionally, we wish to highlight that we have not conducted a quality check of the curves, but rather focused on establishing an overview of the current literature on the curves and the collection of these for the database. When considering their usage, it is essential to also account for the resolution, adaptability, and transferability of the curves in assessing and managing risks to CI across various settings and scenarios. In supporting this, we consistently summarized characteristics of each curve in Table D1 of our database.'*

2. The use of the term "multi-hazard" in the title appears vague, with potential differences between "Multi-hazard" and "multiple hazards." The authors should either redefine "multi" in the title or refocus the article exclusively on multi-hazard events. If opting for the latter, reference to relevant articles (a few suggested below) for clarification on the definition of multi-hazard events is encouraged.

   https://doi.org/10.1016/j.scitotenv.2023.169120
   https://doi.org/10.1038/s43017-020-0060-z
   https://doi.org/10.1002/2013RG000445

*[Authors' reply] Thank you for pointing this out. We have now made sure that the manuscript is clear in its terminology. In our research we exclusively focus on fragility and vulnerability curves for single hazards. We have replaced multi-hazard by either multiple hazards or hazards, including a change of the manuscript's title to 'Review article: Physical Vulnerability Database for Critical Infrastructure Hazard Risk Assessments – A systematic review and data collection'.*

*In addition, we extended our statement in lines 831-834 to communicate that vulnerability research for CI progresses in multi-hazard context, and we also used reviewer's #2 recommended literature for this:*

> *'Moreover, recent studies have also begun to assess the vulnerability of CI due to the joint effect of multiple hazards (e.g., Argyroudis et al., 2018; Teoh et al., 2019; Zhu et al., 2022), aligning with the growing field of multi-hazard research aimed at elucidating the interactions of hazards (e.g., Gill and Malamud, 2014; Lee et al., 2024).'*

*And we have also edited the sixth point (point 5 in original manuscript) in our list of recommendations for the expansion of our database in Section 5 (lines 876-880):*

> *'We strongly encourage users to expand the database with: (1) existing curves that are currently not included, (2) curves for other hazard types, such as wildfires and extreme cold, (3) curves for other important infrastructures types, such as bridges, (4) curves for various building typologies with regard to form (e.g., low-rise) and construction materials, (5) curves that consider the joint effect of multiple intensity measures of a single hazard, and (6) curves that consider the interaction of multi-hazards.'*

3. Referring the statement ""We conducted a literature search for CI vulnerability to flooding, earthquakes, windstorms and landslides over the period January 2022 to March 2023 by systematically using combinations of keywords on the general concept of hazards, critical infrastructure and vulnerability" in section 2.1, methodological concerns include the choice of four specific hazards, the short 14-month (January 2022 to March 2023) search window, and ambiguity regarding whether an established approach like the PRISMA Protocol was followed for the systematic literature search. The authors should specify the databases considered (e.g., Scopus, WoS, PubMed) and provide the search term syntax. Additionally, the number of initial references found, inclusion/exclusion criteria, and handling of duplicate references should be clarified.

*[Authors' reply] The reviewer raises an important concern regarding the methodology applied for our systematic review. We agree that various aspects of our methodology should be clarified to clearly demonstrate that we followed the PRISMA Protocol for the systematic literature search and have made major revisions to do so.*

*Firstly, we included a schematic flow chart to show the methodological framework (new figure 1), including the number of references included/excluded throughout the process of literature search, screening and final selection, which aligns with the established PRISMA protocol. We added the following text to the first paragraph of Section 2.1 (lines 76-77 and lines 83-86):*

*'The schematic workflow for the literature search, screening and final selection of articles for the systematic literature review on the CI vulnerability to flooding, earthquakes, windstorms and landslides is summarized in figure 1.'*

*'The literature search yielded 2,590,003 initial records, gathered from 125 search term syntaxes listed in Appendix A. It became apparent that a substantial number of papers did not address CI vulnerability in context of natural hazards. As a result, we decided to select the first 250 records for each search term syntax, totalling 31,250 records for the screening procedure.'*

*Secondly, we have added 'Appendix A: Search term syntax and number of records' in which we provide an overview of the 125 search term syntax used for our systematic literature review. Additionally, we provide the number of records per search term syntax in Appendix A. Furthermore, we also updated Table 1 that contains the search terms categorized per general concept accordingly. Thirdly, we have also taken care to clarify our methodological framework throughout 'Section 2.1'.*

*With regard to our choice considering the four specific hazards that we include in our research, we added lines 77-79:*

*'The hazards were chosen based on their widespread occurrence, significant potential for damage to CI and historical evidence of their impact on communities.'*

*With regard to our choice of the search window, we would like to highlight that the mentioned 14-month period of January 2022 to March 2023 is not the search window of our systematic review, but the period in which we conducted our systematic literature search. We edited lines 105-106:*

*'However, we did not limit the search window and the geographical scope of the study and are thus still able to provide insight into vulnerability curves in various contexts.'*

*With regard to the search engine, we added lines 80-82 to the first paragraph of section 2.1:*

*'Our review is not restricted to peer-reviewed academic articles as curves are also published in 'grey literature', such as research reports released by governments or engineering firms. We therefore use Google Scholar as search engine that is not limited to academic literature in order to minimize the possibility of excluding relevant information within our research scope.'*

*With regard to the inclusion and exclusion criteria, we have included these in the schematic flow chart of the methodological framework. We now also better indicate the inclusion criteria in the text that were already described in the original manuscript (now lines 91-98). Furthermore, we also added to the description of exclusion criteria in Section 2.1 (lines 100-104).*

*'If multiple records present the same curves, we only include the original source reference. We also excluded records that describe the probability of an asset failing to operate rather than the damage probability of being in a certain physical damage state,*

*as we confine the scope of this research to fragility curves that specifically involve the physical damage (see inclusion criteria 1). Note that we exclude curves at subcomponent level (e.g., circuit switcher), but do include them if they are at asset- or system level.'*

4. Referring the statement "We excluded bridges from our database as there is an excessive amount of bridge literature that deserves a review article on its own" in section 2, the exclusion of bridges from the database lacks a clear rationale, merely stating there is an excessive amount of bridge literature deserving a separate review article. A more explicit justification or reconsideration of this decision is necessary.

*[Authors' reply] Thank you for the useful comment. We added a more explicit justification of our decision to not include bridges. It now reads (lines 116-120):*

> *'We would like to stress that our database does not encompass all types of infrastructure. There is already vast literature available for limited infrastructure types. For bridges, for example, 224 bridge damage curves for 28 primary bridge types are offered by FEMA (2020) and a dedicated review is provided by Muntasir Billah and Shahria Alam (2015). Moreover, retrieving curves is labour-intensive. Instead, our focus was on delivering as comprehensive a review as possible for the infrastructure types as presented in table 1.'*

5. The temporal discrepancy between the search window (January 2022 to March 2023) and data in Figure 1 (1984-2023) raises confusion. The methodology lacks clarity on whether vulnerability functions were extracted from literature or other sources. A comprehensive methodology section, including a flow diagram, is needed to elucidate the study's process.

*[Authors' reply] The time window of January 2022 to March 2023 is the period during which we conducted the literature search and screening. The fragility and vulnerability curves are extracted from the 95 references that can be found in the reference list of our manuscript as well as in the reference list presented in the publicly available database itself. In fact, for each single curve we indicate in the database's 'Table D1 Summary CI Vulnerability Data' the reference from which this information is sourced from. As per reply of the reviewer's comment #3, we have clarified the search window and have added a flow diagram of the methodology to elucidate the study's process.*

6. The unclear methodology and data sources undermine the meaningfulness of the results at this stage.

*[Authors' reply] We have taken care to elucidate the methodology, such as described in reviewer's comments 3-5.*

**Minor comments**

1. In the abstract, the phrase "essential vulnerability information for CI" requires clarification.

*[Authors' reply] We have adjusted the abstract as per reviewer's #1 minor comment on a missing definition of vulnerability in the abstract. We have added a definition for fragility and vulnerability curves for CI for clarification.*

2. The statement, "even within these non-public databases, CI is often limited represented," lacks clarity and needs rephrasing for better understanding.

   *[Authors' reply] As per reviewer's #1 minor comment, we have adjusted this statement.*

---

## Author Comment (AC3)

**Review article: Physical Vulnerability Database for Critical Infrastructure Multi-Hazard Risk Assessments – A systematic review and data collection**
**Authors: Sadhana Nirandjan, Elco E. Koks, Mengqi Ye, Raghav Pant, Kees C.H. Van Ginkel, Jeroen C.J.H. Aerts, and Philip J. Ward**

_**Response to Anonymous Referee #3**_

This is an invaluable study for those engaged in the risk analysis of critical infrastructure for natural hazards. The paper is very well organised and includes a systematic literature review and an extensive database of fragility and vulnerability models, and associated costs for various types of infrastructure, while bridges are excluded due to the excessive amount of bridge literature.

_**[Authors' reply]** We thank the reviewer for the encouraging words with regard scientific novelty and soundness. In the following sections, we address the two valuable comments of the reviewer._

The following comments are provided:

1. It is suggested adding the parameters of the fragility functions when possible (e.g. median, standard deviation when a lognormal distribution is adopted).

   _**[Authors' reply]** Thank you for your suggestion. We have decided to not include the parameters of the curves to the database as they were used to reconstruct the curves if available in the original reference. Instead, we explicitly specify whether parameters are presented in the original reference for users seeking further consultation. Specifically, this information is summarized in the Database's 'Table D1 Summary CI Vulnerability Data' under the column 'Readily available'. Additionally, we use color codes throughout 'Table D2 Hazard Fragility and Vulnerability Curves', which contains the actual curves, to highlight for which curves no parameters were available for the reconstruction of the curves._

2. It is suggested adding in the database the following fragility functions:

Fragility functions for tunnels (earthquakes, and ageing effects):

Argyroudis, S. A., Pitilakis, K. D. (2012). Seismic fragility curves of shallow tunnels in alluvial deposits. Soil Dynamics and Earthquake Engineering, 35, 1-12.

Huang, Z. K., Pitilakis, K., Tsinidis, G., Argyroudis, S., Zhang, D. M. (2020). Seismic vulnerability of circular tunnels in soft soil deposits: The case of Shanghai metropolitan system. Tunnelling and Underground Space Technology, 98, 103341.

Argyroudis, S., Tsinidis, G., Gatti, F., Pitilakis, K. (2017). Effects of SSI and lining corrosion on the seismic vulnerability of shallow circular tunnels. Soil Dynamics and Earthquake Engineering, 98, 244-256

Huang, Z., Argyroudis, S., Zhang, D., Pitilakis, K., Huang, H., Zhang, D. (2022). Time-dependent fragility functions for circular tunnels in soft soils. ASCE-ASME Journal of Risk and Uncertainty in Engineering Systems, Part A: Civil Engineering, 8(3), 04022030.

Tsinidis, G., Karatzetzou, A., Stefanidou, S., Markogiannaki, O. (2022). Developments in seismic vulnerability assessment of tunnels and underground structures. Geotechnics, 2(1), 209-249.

Fragility functions for retaining walls/abutments (earthquakes):

Argyroudis, S., Kaynia, A. M., & Pitilakis, K. (2013). Development of fragility functions for geotechnical constructions: application to cantilever retaining walls. Soil Dynamics and Earthquake Engineering, 50, 106-116.

Fragility functions for embankments (flood, scour)

McKenna, G., Argyroudis, S. A., Winter, M. G., & Mitoulis, S. A. (2021). Multiple hazard fragility analysis for granular highway embankments: Moisture ingress and scour. Transportation Geotechnics, 26, 100431.

Tsubaki, R., Bricker, J. D., Ichii, K., Kawahara, Y. (2016). Development of fragility curves for railway embankment and ballast scour due to overtopping flood flow. Natural Hazards and Earth System Sciences, 16(12), 2455-2472.

> *[Authors' reply] Thank you for sharing these valuable references. As our review also covers road and railway embankment curves, we have decided to add the flood fragility curves for embankments considering scour. Lines 268-274 now reads:*
>
>> *'McKenna et al. (2021) provides analytically derived fragility and vulnerability curves [F7.14-15] for granular highway embankments. They use the Water Intensity Measure (WIM) as intensity measure, which describes the proportion of the embankment height that would be considered saturated if exposed to moisture ingress due to flooding. Additionally, they also assess the impact of scouring using a scouring depth of 0.5 and 3 m as lower and upper boundary, respectively, whilst the raised groundwater level was maintained. Their study shows that higher damages are expected with increasing moisture ingress and scour depths.'*
>
> *And lines 291-296 now read:*
>
>> *'Tsubaki et al. (2016) explains that railway damage commonly occurs due to floodwater overtopping leading to scouring of the ballast and embankment upon which the trail tracks are built. Railway overtopping damage begins with ballast scour and progresses to embankment scour. They therefore developed fragility curves for ballast scour damage, embankment fill scour damage and a combination of both damage conditions [F8.9-11] using damage records of flood events for single-track railways in Japan.'*
>
> *The curves presented by McKenna et al. (2021) and Tsubaki et al. (2016) have also been added to the database. Subsequently, we have updated figures 3 and 4 in the discussion sections 4.1 and 4.2, respectively, as well as the numbers in the main text in these sections. Furthermore, we have adjusted our statement saying that only inundation depth is used as intensity measure for the flood curves (lines 822-825):*
>
>> *'We encounter a range of ground shaking hazard intensity measures for earthquakes such as PGA, PGV and elastic spectral displacement. Conversely, flooding predominantly relies on a*

*single intensity measure, inundation, although there is a rare instance where WIM is used. However, other intensity measures such as flow velocity (Kreibich et al., 2009; Koks et al., 2022) and salinity (Glas et al., 2017) also play an important role to infrastructure damage.'*

---

## Referee Report (RR1)

I thank the authors for their efforts to review my comments. I particularly welcome the inclusion of Figures 1 and 2, which substantially improve the quality of the manuscript. However, there are a few follow up comments below that I think should be addressed before publication. The comment number provided at the start of each comment refers to the previous round of review.

1. Main comment #2a: Suggest changing "at risk to natural hazards" to "exposed and vulnerable to natural hazards"

2. Main comment #2b: The point I was trying to make here is that the output of a vulnerability curve is typically a continuous measure of *loss*, whereas the output of a fragility curve is a measure of *damage*. The authors refer to a damage factor as being the output of a vulnerability curve in line 41, but this is actually a financial loss ratio (as they define in the parenthesis). In other words, the "damage" output of a vulnerability curve is typically expressed in terms of percentage of replacement cost and is therefore actually a loss rather than a damage metric. I would suggest making the distinction between the two types of output more explicit, so that readers have a clear understanding of the differences between the two curves. Along the same lines, the authors should qualify that the "physical damages" captured by the database include consequences in the form of repair cost (i.e., financial loss) ratios (currently referred to as "damage factors") output from vulnerability models.

3. Minor comment #1: "level of susceptibility" seems a little vague to me. I would suggest using something like "level of loss experienced" or "level of impact experienced" to be slightly more specific.

4. Minor comment #2: I don't think the notation "E(C>c)" makes sense when referring to a mean value, which is why I thought you were referring to a probability. I believe it should be re-written as E(C)

5. Minor comment #4: To make this point clearer, I would suggest that you mention the development of both fragility and vulnerability curves being common practice within the earthquake community.

6. Minor comment #6: The statement provided here does not specifically mention that the cost values are country-specific.

7. Minor comment #8: By using the term "building typologies", I think you are still welcoming the possibility of the database being extended beyond the critical infrastructure depicted in Figure 2. This is fine if it is your intention, but, if not, then I would remove the word "building" and use something along the lines of "various forms of critical infrastructure (e.g., in terms of construction material)"

8. Additional comments:
   a. I think that Figure 1 should also account for the "vulnerability" and "critical infrastructure" search terms, in addition to hazards

---

## Author Response (AR2)

**Review article: Physical Vulnerability Database for Critical Infrastructure Multi-Hazard Risk Assessments – A systematic review and data collection**
**Authors: Sadhana Nirandjan, Elco E. Koks, Mengqi Ye, Raghav Pant, Kees C.H. Van Ginkel, Jeroen C.J.H. Aerts, and Philip J. Ward**

_**Response to Anonymous Referee #1**_

I thank the authors for their efforts to review my comments. I particularly welcome the inclusion of Figures 1 and 2, which substantially improve the quality of the manuscript. However, there are a few follow up comments below that I think should be addressed before publication. The comment number provided at the start of each comment refers to the previous round of review.

_**[Authors' reply]** We thank the reviewer for the kind words and for recognizing the improvements made to the manuscript. We greatly appreciate your thorough review and are grateful for your insightful feedback throughout the process. We carefully considered the minor comments provided and addressed them in the final version of the manuscript._

1. Main comment #2a: Suggest changing "at risk to natural hazards" to "exposed and vulnerable to natural hazards"

   _**[Authors' reply]** Thank you for your suggestion. We believe that the phrase 'at risk to natural hazards' effectively communicates the message that we would like to convey. In the sentence that follows, we directly explain why critical infrastructure are increasingly at risk to natural hazards and thus add context to our phrase (lines 26-28):_

   > _'This is driven by both a growing demand for infrastructure associated with socio-economic development, and an observed and projected increase in the frequency and intensity of climate extremes (IPCC, 2022).'_

2. Main comment #2b: The point I was trying to make here is that the output of a vulnerability curve is typically a continuous measure of loss, whereas the output of a fragility curve is a measure of damage. The authors refer to a damage factor as being the output of a vulnerability curve in line 41, but this is actually a financial loss ratio (as they define in the parenthesis). In other words, the "damage" output of a vulnerability curve is typically expressed in terms of percentage of replacement cost and is therefore actually a loss rather than a damage metric. I would suggest making the distinction between the two types of output more explicit, so that readers have a clear understanding of the differences between the two curves. Along the same lines, the authors should qualify that the "physical damages" captured by the database include consequences in the form of repair cost (i.e., financial loss) ratios (currently referred to as "damage factors") output from vulnerability models.

   _**[Authors' reply]** Thank you for clarifying your comment from the first round of revisions. From your perspective, damage should only be used if referred to a physical state of damage and loss should be used when referred to consequences such as the ones expressed in monetary values. We_

*believe that in this sense the type of consequence is an important factor in determining whether one should refer to damages or losses. Damage refers to the physical harm or deterioration suffered by infrastructure or assets (which can be expressed in monetary values), whereas loss encompasses the broader consequences (e.g., increased travel time due to damaged infrastructure).*

*In our manuscript we only review curves that are associated with direct physical asset damages for a selection of hazards. However, we do not use the term 'damage factor' in the remainder of the original manuscript but use the term 'mean damage ratio' which is more commonly used in literature. We believe that introducing another term, such as 'financial loss' as proposed by the reviewer, could lead to confusion; financial losses are often interpreted as indirect economic losses, which are not considered in this study. To address the remark by the reviewer and improve clarity we decided to replace the term 'damage factor' in our introduction and now use 'mean damage ratio' throughout the revised manuscript instead. Lines 37-42 now read:*

> *'These curves relate given levels of a hazard intensity measure (e.g., flood inundation depth, wind speed) to the potential physical damage of an asset. The potential damage can either be expressed in absolute monetary terms, or in relative numbers that are often referred to as the mean damage ratio (MDR), which is commonly expressed as the ratio of the expected repair cost to the replacement costs of a structure (WBG, 2019). In the latter case, the MDR is then multiplied by a cost feature to obtain the potential damage for a given hazard intensity level.'*

3. Minor comment #1: "level of susceptibility" seems a little vague to me. I would suggest using something like "level of loss experienced" or "level of impact experienced" to be slightly more specific.

*[Authors' reply] Thank you for your comment. For consistency with the main text and clarity purposes, we adjusted the definition of vulnerability curves in the abstract to "…quantify the level of damage of an element under varying hazard intensities".*

4. Minor comment #2: I don't think the notation "E(C>c)" makes sense when referring to a mean value, which is why I thought you were referring to a probability. I believe it should be re-written as E(C)

*[Authors' reply] Thanks for your insight. The notation "E(C>c)" was indeed used to refer to a mean value and we have now rewritten this as E(C). This has been adjusted in both the main text under section 2.3 and formula 1.*

5. Minor comment #4: To make this point clearer, I would suggest that you mention the development of both fragility and vulnerability curves being common practice within the earthquake community.

*[Authors' reply] Whilst we agree that the earthquake community uses both fragility and vulnerability curves, as does the flooding community, the main purpose of the phrase is to emphasize that we do see a difference in focus in both the communities, which is also backed up by literature for reference purposes. To improve clarity, we have now adjusted our phrase (lines 44-46):*

> *'The development of fragility curves is particularly emphasized within the earthquake community (Douglas, 2007), whereas the flood community tends to focus more on the development of vulnerability curves (Meyer et al., 2013).'*

*This wording suggests that while fragility curves are prominent in earthquake studies, vulnerability curves are more commonly emphasized in flood risk assessments, without implying that one community does not use the other approach.*

6. Minor comment #6: The statement provided here does not specifically mention that the cost values are country-specific.

*[Authors' reply] Thank you for your comment. We specify in the statement provided that the cost-values are specific to certain country/countries. We refer to this as 'the geographical application', and this term is also consistently used throughout the database itself.*

7. Minor comment #8: By using the term "building typologies", I think you are still welcoming the possibility of the database being extended beyond the critical infrastructure depicted in Figure 2. This is fine if it is your intention, but, if not, then I would remove the word "building" and use something along the lines of "various forms of critical infrastructure (e.g., in terms of construction material)"

*[Authors' reply] Thank you for pointing this out. Our original intention was to suggest that our database could be further enriched with curves that represent various building typologies with regard to form and construction materials as these play an important role in the level of vulnerability to a hazard. For example, a high-rise hospital may be more vulnerable to windstorms compared to low-rise hospitals, and concrete schools are more vulnerable to earthquakes compared to schools with steel-reinforced concrete. We believe that the database should be expanded with curves that better capture these variations in vulnerability due to these kind of characteristics. However, this not only applies to buildings (e.g., health and educational facilities) but to other critical infrastructure types as well. Based on your suggestion, we now made a slight adjustment to our wording to now read (lines 876-877):*

> *'We strongly encourage users to expand the database with: [...] (4) curves for various infrastructure characteristics, such as form (e.g., low-rise) and construction materials'*

8. Additional comments:  a. I think that Figure 1 should also account for the "vulnerability" and "critical infrastructure" search terms, in addition to hazards

*[Authors' reply] Thank you for your comment. The number of records found for the 125 search term syntaxes are listed in Appendix A. This appendix clearly demonstrates the number of records found for the 'vulnerability' and 'critical infrastructure' search terms. In figure 1, we decided to show the 'number of records removed' categorized per hazard to be aligned with sections 3.1-3.4 which are also organized per hazard type. Moreover, the keywords that we used for the "vulnerability" and "critical infrastructure concepts" for the literature search are presented in Table 1. We believe that we provide sufficient information about the search terms syntaxes and the number of records in the main text and the figures, and that the reader should refer to Appendix A for more detailed information regarding the number of records found for the 'vulnerability' and 'critical infrastructure' search terms.*

***Response to Anonymous Referee #3***

The authors have done an excellent job revising the manuscript. A few minor comments are provided for the authors to consider in the final version.

*[Authors' reply] We thank the reviewer for the positive remarks and are pleased that he/she recommends publication in Natural Hazards and Earth System Sciences following a revision of technical corrections. We have addressed these points in the revised manuscript and agree that this has led to a further refinement of the manuscript.*

1. It is suggested to review the definition of vulnerability curves in the abstract "...quantify the level of susceptibility of an element under varying hazard intensities"; it is not clear how susceptibility is defined here - and make it consistent with the definition given in the main text "These curves relate given levels of a hazard intensity measure to the potential physical damage of an asset".

*[Authors' reply] Thank you for your comment. For consistency with the main text and clarity purposes, we adjusted the definition of vulnerability curves in the abstract to "...quantify the level of damage of an element under varying hazard intensities".*

2. It is suggested to include a generic example (figure) of a fragility and vulnerability curve.

*[Authors' reply] Thank you for your suggestion. We have now added a generic example of fragility and vulnerability curves to our manuscript as figure 2 under section 2.3 and adjusted the numbering of the other figures accordingly.*

3. An alternative version of the title is the following: "Physical Vulnerability of Critical Infrastructure: A Systematic Review and Database for Hazard Risk Assessment"

*[Authors' reply] Thank you for your suggestion regarding the title of the manuscript. After careful consideration, we believe that the current title 'Physical Vulnerability Database for Critical Infrastructure Hazard Risk Assessments – A systematic review and data collection' accurately reflects the focus and content of the manuscript. We included the 'data collection' aspect to our title because it is integral to our work and highlights the significant effort put into compiling and organizing the data.*

4. Check the references, eg the following is a repetition:

Kakderi, K. and Argyroudis, S.: Fragility Functions of Water and Waste-Water Systems, in: Geotechnical, Geological and
Earthquake Engineering, vol. 27, edited by: Pitilakis, K., Crowley, H., and Kaynia, A., Springer, Dordrecht, 221–258,
1060 https://doi.org/10.1007/978-94-007-7872-6_8, 2014a.

Kakderi, K. and Argyroudis, S. A.: Chapter 8 Fragility Functions of Water and Waste-Water Systems, in: Fragility Functions of Water and Waste-Water Systems, https://doi.org/10.1007/978-94-007-7872-6, 2014b.

*[Authors' reply] Thank you for pointing this out. This is indeed a repetition and we have now corrected this in both the main text and the reference list.*